# Role of BRCA2 DNA-binding and C-terminal domain in its mobility and conformation in DNA repair

Maarten W Paul[1†], Arshdeep Sidhu[1,2†‡], Yongxin Liang[1],
Sarah E van Rossum-Fikkert[1,2], Hanny Odijk[1], Alex N Zelensky[1], Roland Kanaar[1],
Claire Wyman[1,2]*

[1]Department of Molecular Genetics, Oncode Institute, Erasmus MC Cancer Institute, Erasmus University Medical Center, Rotterdam, Netherlands; [2]Department of Radiation Oncology, Erasmus University Medical Center, Rotterdam, Netherlands

**\*For correspondence:**
c.wyman@erasmusmc.nl

[†]These authors contributed equally to this work

**Present address:** [‡]Arshdeep Sidhu, Division of Molecular Genetics and Cancer, Nitte University Centre for Science Education and Research, Nitte (Deemed to be University), Mangalore, India

**Competing interests:** The authors declare that no competing interests exist.

**Abstract** Breast cancer type two susceptibility protein (BRCA2) is an essential protein in genome maintenance, homologous recombination (HR), and replication fork protection. Its function includes multiple interaction partners and requires timely localization to relevant sites in the nucleus. We investigated the importance of the highly conserved DNA-binding domain (DBD) and C-terminal domain (CTD) of BRCA2. We generated BRCA2 variants missing one or both domains in mouse embryonic stem (ES) cells and defined their contribution in HR function and dynamic localization in the nucleus, by single-particle tracking of BRCA2 mobility. Changes in molecular architecture of BRCA2 induced by binding partners of purified BRCA2 were determined by scanning force microscopy. BRCA2 mobility and DNA-damage-induced increase in the immobile fraction were largely unaffected by C-terminal deletions. The purified proteins missing CTD and/or DBD were defective in architectural changes correlating with reduced HR function in cells. These results emphasize BRCA2 activity at sites of damage beyond promoting RAD51 delivery.

## Introduction

Breast cancer type two susceptibility protein (BRCA2) is a required component in multi-step genome maintenance processes that are coordinated in time and place. BRCA2 knock-out is lethal in mammalian cells, and a defective BRCA2 causes increased sensitivity to genotoxic agents, defective DNA repair, and reduced homologous recombination (HR) activity (*Prakash et al., 2015*; *Sharan et al., 1997*; *Yu et al., 2000*). One role of BRCA2 common to DNA break repair, DNA crosslink repair, and replication fork protection is delivery of RAD51 to sites where it is needed (*Holloman, 2011*; *Sharan et al., 1997*; *Yuan et al., 1999*). RAD51 is also an essential protein whose biochemical function is to form filaments on single-stranded DNA (ssDNA) capable of performing strand exchange reactions with homologous partners or otherwise protecting the bound DNA (*Baumann and West, 1998*; *Heyer et al., 2010*). We consider essential BRCA2 activity to involve at least (1) spatial relocation in the nucleus resulting in accumulation at sites where RAD51 is needed and (2) molecular rearrangement to release or deposit RAD51 on DNA in an active form.

Accumulation of the required proteins at the sites of DNA damage is typically defined as the appearance of foci, high local concentration of proteins, in immunofluorescence experiments. During HR, the formation of RAD51 foci is considered a critical step, and this has recently also been introduced in clinical settings as a test for HR defects in tumors (*Naipal et al., 2014*). The accumulation of RAD51 into foci depends on functional BRCA2 (*Yuan et al., 1999*). Several studies have addressed the role of different interactors and domains of BRCA2 in foci formation after DNA-damage induction (*Shahid et al., 2014*). The presence of Partner and Localizer of BRCA2 (PALB2) and its

interaction with the N-terminus of BRCA2 are essential for the localization of BRCA2 and RAD51 to foci (*Oliver et al., 2009*; *Xia et al., 2007*; *Xia et al., 2006*), whereas the loss of interaction affects HR and genome stability in general (*Hartford et al., 2016*). In chicken DT40 cells, both N-terminal interaction with PALB2 and C-terminal DNA-binding domain (DBD) have a role in focal accumulation of BRCA2; accumulation is fully eliminated if neither domain is present (*Al Abo et al., 2014*). Additionally, the interaction of the BRCA2 DBD with the small DSS1 protein is required for proper localization of BRCA2 to the nucleus and for BRCA2 and RAD51 focus formation (*Gudmundsdottir et al., 2004*; *Kojic et al., 2005*; *Li et al., 2006*).

Accumulation of BRCA2 and RAD51 in DNA-damage-induced foci necessarily requires a change in their diffusive behavior. Single-particle tracking (SPT) in living mouse embryonic stem (mES) cells revealed that BRCA2 diffuses as multimeric complexes bound to all detectable nuclear RAD51 (*Reuter et al., 2014*). Individual BRCA2 particles diffuse slowly and are transiently immobile, and this immobility increases in response to DNA-damage induction (*Reuter et al., 2014*). Although they diffuse together, BRCA2 and RAD51 are separated at sites where they accumulate, as determined by super-resolution microscopy (*Sánchez et al., 2017*; *Whelan et al., 2018*). This suggests structural rearrangements of the complex to release RAD51. Purified BRCA2 protein shows remarkable rearrangement by RAD51 and ssDNA (*Le et al., 2020*; *Sánchez et al., 2017*; *Sidhu et al., 2020*). This apparent structural plasticity is a hallmark of proteins with intrinsically disordered regions (*Dunker et al., 2005*; *Gunasekaran et al., 2003*; *van der Lee et al., 2014*), which we hypothesize could be relevant for BRCA2 function in cells.

BRCA2 has many predicted disordered regions, which complicate understanding the functional organization and possible dynamic rearrangement of the reported structured domains. Several crystal structures of small fragments of BRCA2 are available: C-terminal BRCA2-DSS1 complex (*Yang et al., 2002*) (PDB ID: 1IYJ), N-terminal BRCA2-PALB2 (*Oliver et al., 2009*) (PDB ID: 3EU7), Brc4 BRCA2-RAD51 (*Pellegrini et al., 2002*) (PDB ID: 1N0W), BRCA2 phosphopeptide-Plk1 (*Ehlén et al., 2020*) (PDB ID: 6GY2), Brc8-2 BRCA2-RadA (*Lindenburg et al., 2020*) (PDB ID: 6HQU), and a BRCA2 peptide residing in exon 12 with the C-terminal armadillo-type domain of HSF2BP (*Ghouil et al., 2020*). Of these, the largest crystallized segment of BRCA2 is the DBD (736 amino acids), which encompasses the helical domain, tower domain, and the three oligonucleotide/oligosaccharide binding (OB) folds. There are two distinct regions of BRCA2 that bind RAD51 with different consequences (*Esashi et al., 2005*; *Galkin et al., 2005*). The eight centrally located BRC repeats bind multiple RAD51 molecules (*Jensen et al., 2010*). The BRC repeat-RAD51 interactions are important for localizing RAD51 into nuclear foci (*Chen et al., 1999*) and promoting RAD51 filament nucleation in vitro (*Shahid et al., 2014*). There is another RAD51-interaction domain at the C-terminus of BRCA2, which is inhibited by cell-cycle-regulated BRCA2 phosphorylation (*Esashi et al., 2005*). This region, the C-terminal domain (CTD) of BRCA2, which is equivalent to exon 27, has a specific role in protecting replication forks (*Feng and Jasin, 2017*; *Lomonosov et al., 2003*; *Schlacher et al., 2011*). The C-terminal RAD51-binding site is also suggested to stimulate RAD51-mediated recombination and stabilize RAD51 filaments (*Esashi et al., 2005*) while the DBD in combination with DSS1 is suggested to exchange Replication protein A (RPA) for RAD51 on ssDNA (*Yang et al., 2002*; *Zhao et al., 2015*). However, the DBD is not essential for cells to survive (*Edwards et al., 2008*), and although it is the most conserved part of BRCA2 (*Yang et al., 2002*), its exact cellular function remains elusive.

It is essential to understand how BRCA2 works to know how the regions of BRCA2, interacting with multiple partners, contribute to its function. Here we consider the influence of DBD and CTD on dynamic activities of BRCA2, DNA-damage-induced changes in diffusion and partner-binding-induced changes in protein conformation. To understand which parts of BRCA2 are responsible for dynamic localization and structural transitions, we correlated the cellular phenotypes, diffusion dynamics, and in vitro structural transitions for BRCA2 variants lacking either the DBD, the CTD, or both. We discovered that the separate domains do not play a significant role in BRCA2's nuclear localization and diffusion dynamics or RAD51 accumulation but strongly affect protein conformational response to binding partners. The BRCA2 conformational changes correlate with cellular HR activity. Here we discuss the possible importance of BRCA2 for steps in the HR beyond its identified role in RAD51 delivery.

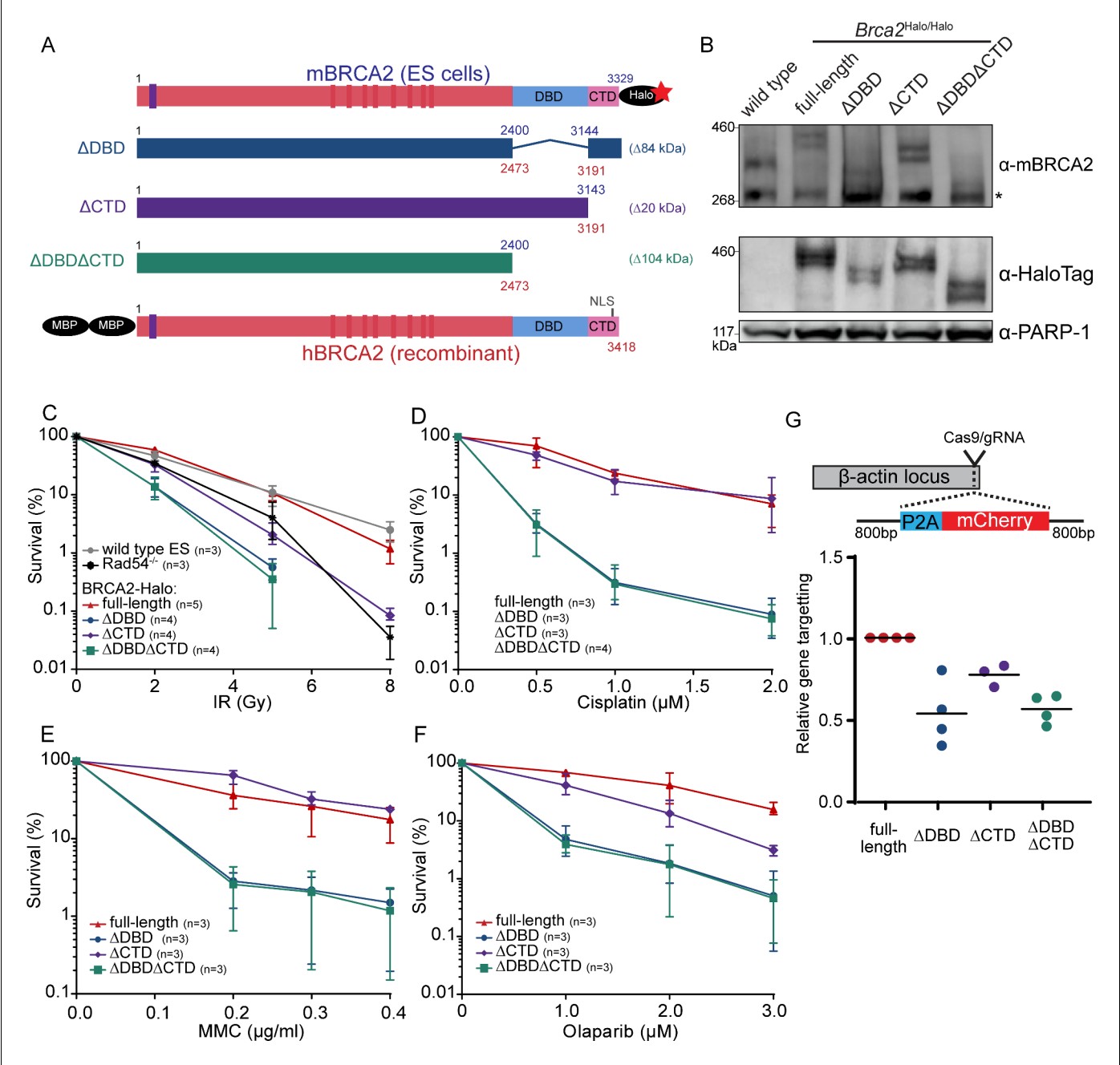

**Figure 1.** Functional analysis of BRCA2 deletion variants in mouse embryonic stem cells, tagged at the endogenous locus with a HaloTag. (**A**) Schematic overview of full-length mouse (top) and human (bottom) BRCA2 proteins, with key domains (DBD, CTD, NLS, BRC1-8: red bars; PALB2-binding: blue bar) and tags indicated. Deletion variants are shown in the middle. Amino acid numbers are shown in blue (mouse) and red (human). Expected molecular weight decrease for the deletion variants is shown on the right. Sequence conservation and alignment between mouse and human BRCA2 DNA-binding domain (DBD) and C-terminal domain (CTD) can be found in *Figure 1—figure supplement 3*. (**B**) Immunoblot of total protein extract from mouse embryonic stem (mES) cells probed with indicated antibodies. Asterisk shows a specific band. Validation of the cell lines by genotyping is described in *Figure 1—figure supplement 1*. Images of the full blots are shown in *Figure 1—source data 1*. (**C–F**) Clonogenetic survivals after ionizing radiation (IR), olaparib, mitomycin C (MMC), and cisplatin treatment with the indicated doses. At 8 Gy of IR, the percentage of surviving colonies of the ΔDBD- and ΔDBDΔCTD-Halo was too low to accurately determine the survival. Error bars indicate the range of data points. n numbers in the figure indicate the number of technical replicates executed on different days. Source data and statistics are available in *Figure 1—source data 1*. (**G**) CRISPR/Cas9-based homologous recombination assay to assess the homologous recombination proficiency of the different BRCA2 mutants. mES cells were transfected with a plasmid encoding Cas9 and the specific guide RNA (gRNA) and a repair template with the self-cleaving peptide P2A and the mCherry sequence in between two homology arms. Upon proper integration of the donor sequence at the ß-actin locus, the cells

*Figure 1 continued on next page*

*Figure 1 continued*

expressed mCherry. 96 hr after transfection, cells were sorted and the frequency of mCherry-positive cells was measured (*Figure 1—figure supplement 2*, *Figure 1—source data 1*). To correct for the difference in transfection efficiency, a plasmid expressing blue fluorescent protein (BFP2) was co-transfected. The frequency of positive cells in every experimental replicate is normalized against wild-type BRCA2-Halo cells. Every data point indicates a technical replicate (averaged from two transfections). p-values (paired two-sided t-test) compared to full-length for the deletion variants are p = 0.0186 (ΔDBD), p = 0.0291 (ΔCTD), and p = 0.0021 (ΔDBDΔCTD), respectively.

The online version of this article includes the following source data and figure supplement(s) for figure 1:

**Source data 1.** Excel file with the source data of the cell survival data in *Figure 1C–F* and HR assay in *Figure 1G*.
**Source data 2.** Full western blot images from *Figure 1B*.
**Figure supplement 1.** Generation of different variants of BRCA2-HaloTag knock-ins.
**Figure supplement 2.** Flow cytometry analysis of the CRISPR/Cas9 β-actin-P2A-mCherry targeting assay as shown in *Figure 1G*.
**Figure supplement 3.** Sequence alignment between human and mouse BRCA2 DBD and CTD.

## Results

To determine the role of DBD and CTD in DNA-damage repair, BRCA2 mobility, and structural plasticity, we created murine cell lines and purified human BRCA2 protein lacking these domains (*Figure 1A*). mES lines producing tagged variants of BRCA2 (full-length; ΔDBD, containing an internal deletion of amino acids 2401–3143; ΔCTD, truncated at 3143; and ΔDBDΔCTD, truncated at 2401) were engineered by homozygous modification of the endogenous *Brca2* alleles and addition of a HaloTag at the end of the coding sequence (*Figure 1B*, *Figure 1—figure supplement 1*). This allowed us to visualize BRCA2 in live or fixed cells (*Los et al., 2008*) and study the effect of deletions under native expression in the absence of wild-type BRCA2 (*Figure 2A*, *Figure 3*). To analyze the role of DBD and CTD in vitro, we purified variants of human BRCA2 protein containing the same deletions (*Figure 1A*, *Figures 4* and *5*).

### Loss of BRCA2 DBD and CTD impairs cell survival and gene targeting

Under unperturbed conditions, the generated cell lines did not show obvious defects in their growth rate. To investigate whether the loss of DBD or CTD affected sensitivity of the cells to DNA damage, we performed clonogenic survival assays after treatment with different DNA-damaging agents: double-strand break induction by ionizing radiation (IR), replication disruption by PARP inhibitor - olaparib, and DNA crosslink induction by mitomycin C (MMC) and cisplatin. Loss of CTD caused sensitization to IR (at 5 Gy up to fivefold decrease in surviving fraction, p<0.001), comparable to the effect of deleting the non-essential auxiliary HR protein RAD54 (*Essers et al., 1997*). In contrast, deletion of DBD further increased sensitization to IR (18-fold decrease in surviving fraction [p<0.001] or 3.6-fold more than in ΔCTD), which was not further exacerbated if CTD was also missing (p<0.001) (*Figure 1C*). Whereas IR induces double-strand breaks directly, PARP inhibitors prevent repair of single-strand breaks that result in the formation of double-strand breaks. The effects of domain deletion on olaparib sensitivity were similar to their effects on IR sensitivity. Cisplatin and MMC induce interstrand crosslinks, which are resolved by mediation of HR proteins such as BRCA2. Different from the results with IR and olaparib, loss of CTD did not lead to a significant sensitization to cisplatin (p = 0.80) nor MMC (p = 0.11) (*Figure 1D–F*). Together, these results indicate that the BRCA2 DBD is important for efficient HR-mediated DNA repair while the CTD is less critical for this cellular activity.

To assay homology search and DNA strand exchange functions of HR, we performed a fluorescence-activated cell sorting (FACS)-based gene-targeting assay (*Yao et al., 2017*), in which Cas9 is used to induce a double-strand break in the β-actin locus that is repaired by a donor plasmid including an mCherry coding sequence (*Figure 1G*). The absolute gene-targeting frequency of about 5% (*Figure 1—figure supplement 2*, *Figure 1—source data 1*) was reduced twofold in BRCA2 ΔDBD (p = 0.0186) and ΔDBDΔCTD (p = 0.0021) cells, while CTD deletion caused an intermediate effect (p = 0.0291). Thus, DNA-damage sensitivity described above correlates with this gene-targeting assay, indicating that BRCA2 DBD is specifically important for HR activity at two-ended DNA breaks.

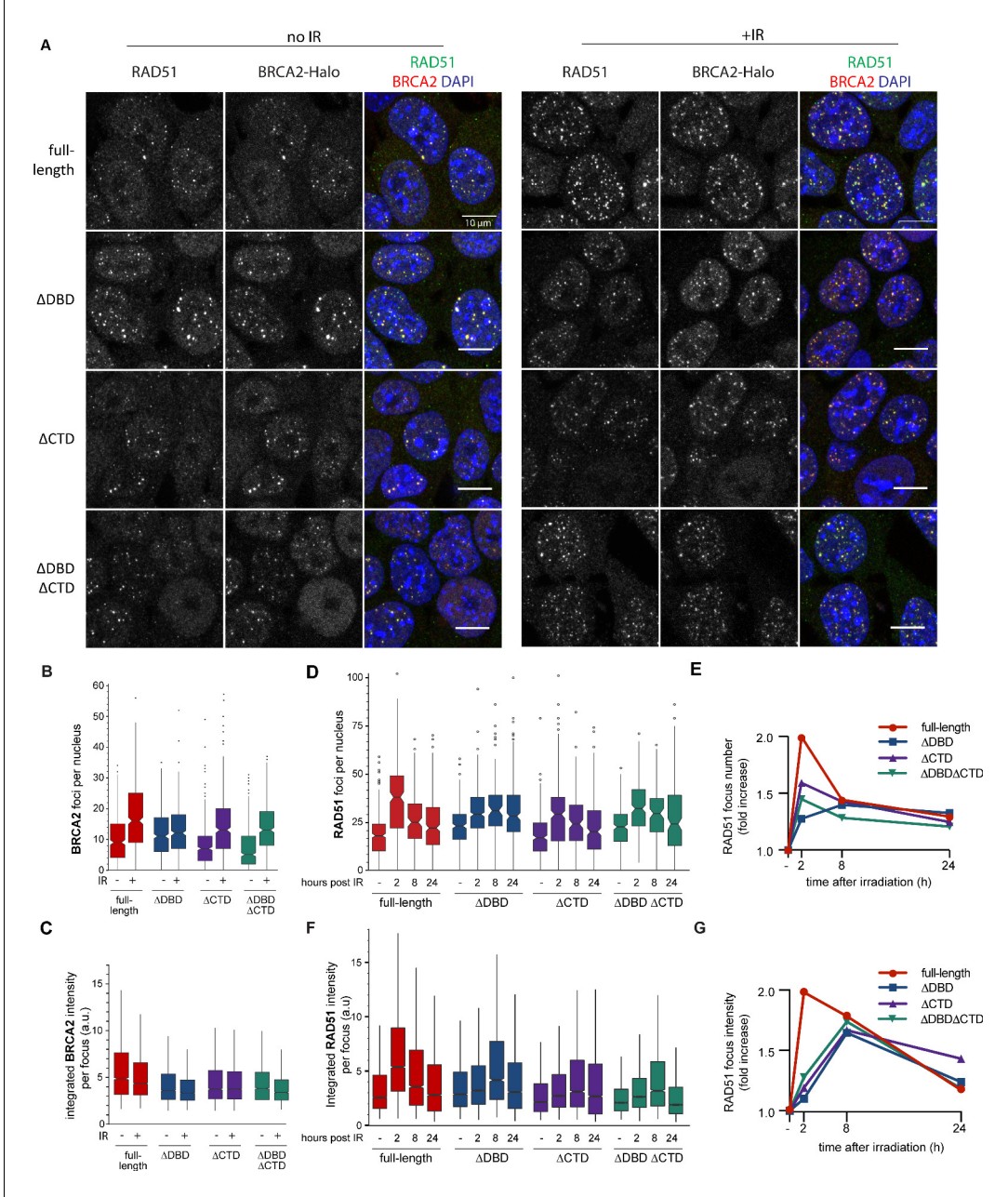

**Figure 2.** BRCA2 and RAD51 foci quantification. (**A**) Representative confocal images (maximum intensity projections) of BRCA2 (red) and RAD51 (green) foci in mouse embryonic stem (mES) cells fixed 2 hr after mock or 2 Gy irradiation, without pre-extraction. Scale bar, 10 μm (full images can be found at *Figure 2—source data 2*). (**B**) Quantification of the number of BRCA2-Halo (JF646) foci per nucleus of EdU+ cells irradiated with 2Gy ionizing radiation (IR) in cells without pre-extraction; three technical replicates, at least 250 cells per condition (*Figure 2—source data 1*). (**C**) Distribution of integrated BRCA2 intensity per focus. (**D**) Quantification of the number of RAD51 foci in EdU+ cells irradiated with 2 Gy IR and fixed after indicated number of hours with pre-extraction for RAD51 immunostaining. Example images and percentage of EdU+ cells per condition are shown in *Figure 2— figure supplement 1*; three technical replicates, at least 100 cells per condition (statistical data available in *Figure 2—source data 1*). (**E**) Fold change of foci number with respect to untreated cells. (**F**) Integrated RAD51 intensity per focus. (**G**) Fold change in integrated intensity of RAD51 foci relative to untreated cells. Representative images are shown in *Figure 1—figure supplement 1*. Data plotted per time point can be found in *Figure 2—figure supplement 1C,E*. In boxplots in (**C**) and (**F**), distribution outliers are not shown; source data is available in *Figure 2—source data 1*.

The online version of this article includes the following source data and figure supplement(s) for figure 2:

**Source data 1.** Excel file with exact n numbers and statistical tests of *Figure 2B–G* and the source data of the foci quantification.
**Source data 2.** Original uncropped images from *Figure 2A*.
**Figure supplement 1.** Additional images and plots for BRCA2 and RAD51 foci quantification.

## DBD and CTD affect BRCA2 and RAD51 focus kinetics

A critical initial step of HR in cells involves BRCA2-mediated RAD51 localization to nuclear sites where it is needed, typically observed as foci in cell imaging. We focused on the response of BRCA2 and RAD51 to IR-induced DNA damage because timing of this response in wild-type mES cell lines is well defined, and in contrast to genotoxic chemicals, damage induction is instantaneous and synchronous. We visualized BRCA2 protein with a bright photostable fluorophore via the HaloTag using JF646 HaloTag ligand (*Grimm et al., 2015*) combined with RAD51 immunofluorescence (*Figure 2A*). As the DBD of BRCA2 binds DNA in vitro (*Yang et al., 2002*), it might contribute to BRCA2 localization and/or retention at the sites of damage. However, we observed formation of both spontaneous and IR-induced nuclear BRCA2 and RAD51 foci in all three BRCA2 deletion variants, where BRCA2 and RAD51 foci appeared to overlap to a large extent (*Figure 2A*).

## DBD and CTD affect the amount of RAD51 and BRCA2 at repair sites

Absence of a clear qualitative effect on foci formation was unexpected, so we performed further systematic quantification of fluorescence of BRCA2-Halo-JF646 and RAD51, by immunofluorescence, in fixed cells. Only the CTD deletion affected BRCA2 foci, and in the absence of induced DNA damage (background), there was a reduction in their number compared to full-length BRCA2 (20% reduction, p<0.001) (*Figure 2B*). Upon irradiation, the number of BRCA2 foci increased in all deletion variants, (p<0.001 for all BRCA2 variants). However, total number of BRCA2 foci appeared lower than full-length after radiation in all deletion variants. The effect of DBD and CTD deletion on the intensity of background BRCA2 foci was much more pronounced (1.5-fold reduction, p<0.001) (*Figure 2C*). Interestingly, in all cell lines, IR-induced increase in the number of foci was accompanied by a decrease in focus intensity, suggesting that BRCA2 re-localizes from the background to the IR-induced foci, but this effect was suppressed in the deletion variants (only 13% reduction in ΔDBD compared to 26% in cells expressing full-length BRCA2).

We further analyzed RAD51 focus formation and resolution over 24 hr after IR treatment (*Figure 2D,E*, *Figure 2—figure supplement 1*). As with BRCA2 foci, the number of RAD51 foci increased in both deletion variants and the control cells (p<0.001 for all BRCA2 variants). Consistent with our previous observations in wild-type ES cells, the number of RAD51 foci peaked 2 hr after IR, then gradually decreased over time reaching near-background levels at 24 hr (*Figure 2D,E*). For the ΔDBD cells, the number of foci increased but did not decrease over time, remaining high at 24 hr.

The return to background number of foci was suppressed to a lesser extent in ΔCTD and the double mutant. The effect of DBD and CTD deletion on RAD51 focus intensity dynamics was also pronounced. In the control cells, changes in RAD51 foci intensity paralleled changes in their number: peaking at 2 hr, decreasing gradually thereafter (*Figure 2F*). In all the three deletion variants, focus intensity increase was reduced (1.2-fold increase compared to 2-fold in control) or delayed (peak at 8 hr compared to 2 hr in control) (*Figure 2F,G*). Taken together, these results show that the deletion mutants of BRCA2 do accumulate RAD51 proteins to IR-induced lesions; however, less RAD51 accumulates and its turnover is supressed.

## DBD and CTD are not essential for BRCA2 mobility response to DNA damage

Previously we used SPT to determine the diffusive behavior of BRCA2-GFP and observed a mobile fraction, which diffused slower than expected due to frequent transient interactions, and an immobile fraction, which increased upon induction of DNA damage (*Reuter et al., 2014*). Interaction between DBD and DNA could be responsible for both restricting the diffusion of the mobile BRCA2 complexes and immobilization upon DNA damage. To test this hypothesis, we performed SPT analysis of BRCA2 deletion variants labeled with JF549 via the HaloTag. The increased photostability and brightness of the JF549 fluorophore compared to green fluorescent protein (GFP) allowed us to follow the mobility for extended periods of time and at an increased frame rate (2000 vs 200 frames, at 33 vs 20 fps, for BRCA2-HaloTag-JF549 and -GFP, respectively). To identify cells in S-phase, we used Proliferating cell nuclear antigen (PCNA) fused with iRFP720 (*Figure 3A*). We tracked several hundred BRCA2 particles per nucleus where individual tracks appear as mobile or immobile (*Figure 3B*, *Figure 3—videos 1* and *2*) and sometimes switch behavior. The diffusive behavior of BRCA2 was quantified using a recently developed deep-learning algorithm to segment tracks into parts

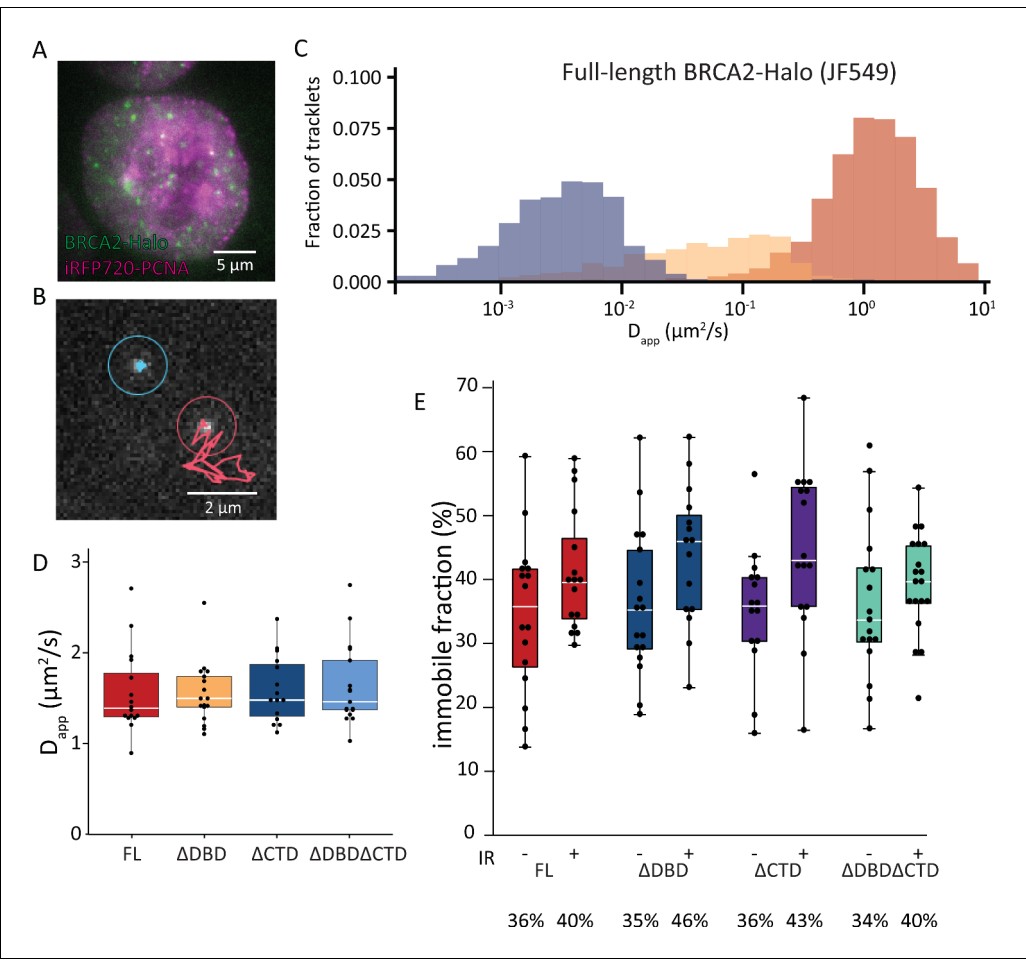

**Figure 3.** Single-particle tracking of BRCA2-HaloTag reveals immobilization of BRCA2 lacking either DBD or CTD upon DNA damage. (**A**) Wide-field image of an S-phase cell visualized with iRFP720-PCNA and BRCA2-HaloTag:: JF549. (**B**) Example of two tracks of BRCA2-Halo showing different diffusive behavior; see also *Figure 3—videos 1–4*. (**C**) Distribution of apparent diffusion coefficients of segmented tracks (tracklets) for immobile (blue), slow (yellow), and fast (red) molecules for full-length BRCA2 in untreated cells; plots for ionizing radiation (IR)-treated cells and other BRCA2 variants are shown in *Figure 3—figure supplement 1*. (**D**) Apparent diffusion rate of fast diffusing BRCA2 tracklets for full-length BRCA2 and indicated deletion variants. p-values (two-sided t-test) comparing full-length with deletion variants (ΔDBD, ΔCTD, ΔDBDΔCTD) are, respectively, p = 0.953, p = 0.797, p = 0.593. (**E**) Immobile fraction estimated by segmentation of tracks by their immobile, slow, or fast mobility (tracklets). Fraction is defined as the percentage of tracklets per cell that are immobile. Cells were imaged between 2 and 4 hr after IR treatment. p-values (two-sided t-test) comparing -/+ IR for different variants (full-length, ΔDBD, ΔCTD, ΔDBDΔCTD) are, respectively, p = 0.08, p = 0.057, p = 0.02, p = 0.4. Merged data from two independent experiments of at least 15 cells and about 10,000 tracks per condition are shown (*Figure 3—source data 1*). Percentages below the plot indicate the median immobile fraction of tracklets per condition.

The online version of this article includes the following video, source data, and figure supplement(s) for figure 3:

**Source data 1.** Excel file with exact n numbers and statistical tests of *Figure 3* and the source data of the single-molecule-tracking experiments.

**Figure supplement 1.** Diffusion histograms of single-particle tracking experiments.

**Figure 3—video 1.** Single-molecule recording of full-length BRCA2-Halo (JF549) in untreated mouse ES cells imaged at 30 ms interval.

https://elifesciences.org/articles/67926#fig3video1

**Figure 3—video 2.** Single-molecule recording of full-length BRCA2-Halo (JF549) in mouse ES cells treated with 2Gy of ionizing radiation imaged at 30 ms interval.

https://elifesciences.org/articles/67926#fig3video2

**Figure 3—video 3.** Single-molecule recording of BRCA2 ΔDBDΔCTD-Halo (JF549) in untreated mouse ES cells imaged at 30 ms interval.

*Figure 3 continued on next page*

*Figure 3 continued*

https://elifesciences.org/articles/67926#fig3video3

**Figure 3—video 4.** Single-molecule recording of BRCA2 ΔDBDΔCTD-Halo (JF549) in mouse ES cells treated with 2Gy of ionizing radiation imaged at 30 ms interval.

https://elifesciences.org/articles/67926#fig3video4

(tracklets) with different mobile states (*Arts et al., 2019a*). An apparent diffusion constant was extracted for each class of segmented tracklets. This revealed different populations of BRCA2 molecules (*Figure 3C*), one with a low apparent diffusion coefficient between 0.001 and 0.01 μm²/s, which we considered immobile, a second fraction of slow mobile molecules with an apparent diffusion coefficient between 0.01 and 0.1 μm²/s, and a third fraction of mobile molecules with an average diffusion rate of 1.5 μm²/s — consistent with our previous results tracking BRCA2-GFP in mES cells (*Reuter et al., 2014*). Also, comparable to our previous work, the fraction of immobile molecules, ~34% in untreated cells (*Figure 3C*), increased to 41% after DNA-damage induction by IR (*Figure 3E*).

The BRCA2 deletion variants all had a similar apparent diffusion coefficient; mobile molecules diffuse with a rate similar to the full-length protein (*Figure 3D*, *Figure 3—figure supplement 1*, *Figure 3—videos 3* and *4*). The increase in immobile tracklets after IR for the variants BRCA2 ΔDBD and ΔCTD was similar to full-length, indicating that these domains separately are not essential for this change in mobility (*Figure 3E*). However, the immobile fraction for BRCA2 ΔDBDΔCTD, missing both regions, did not increase after IR as much as the others (3% increase compared to 7–10%; *Figure 3D*). Thus, either DBD or CTD is sufficient for BRCA2 mobility changes in response to IR but a protein missing both of these domains reduces this response. As deletion of single domains, ΔDBD or ΔCTD, did cause increased sensitivity to DNA-damaging agents (*Figure 1*), we can conclude that diffusion changes in response to DNA damage were not sufficient to assure cell survival or proper HR activity.

## Architectural rearrangement of BRCA2 variants

Our observations so far indicated that BRCA2 function needed for DNA-damage survival and HR includes activities beyond immobility. We considered that HR DNA-damage response requires dynamic interaction between BRCA2 and RAD51 at a scale not evident in our (live) cell imaging. Although BRCA2 and RAD51 diffuse together in the nucleus, they are separated at the sites of DNA damage, requiring a local change in RAD51 and BRCA2 interaction (*Reuter et al., 2014*; *Sánchez et al., 2017*). Previously, we have defined distinct architectural changes in full-length BRCA2 upon association with RAD51 and ssDNA as evidence for such a dynamic interaction (*Sánchez et al., 2017*; *Sidhu et al., 2020*). To correlate BRCA2 architectural changes with in vivo functions, we purified variants of human BRCA2 and deletion variants analogous to those tested in mES cells (*Figure 1A*, *Figure 4—figure supplement 4*). Scanning force microscopy (SFM) imaging revealed that purified BRCA2 exists as a mixture of particles of varying size (multimeric form) and shape (compact to extended). As described in the section below, these features were quantified for all individual proteins/complexes from SFM images by measuring volume and solidity (*Sánchez et al., 2017*; *Sidhu et al., 2020*). The architectural plasticity of full-length BRCA2 is evident in the change in distribution of these features upon addition of binding partners, RAD51 or ssDNA oligonucleotides (*Sánchez et al., 2017*; *Sidhu et al., 2020*).

## CTD and DBD contribute to BRCA2 self-oligomerization

All three BRCA2 deletion variants exist as a distribution of irregular oligomeric molecules as previously observed for the full-length protein (*Figure 4A,B* and *Figure 4—figure supplement 1*; *Sánchez et al., 2017*; *Sidhu et al., 2020*). In the conditions used here, the majority (70%) of full-length BRCA2 was present as assemblies larger than tetramers (*Figure 4B*, left panel). The C-terminal deletion variants showed reduced oligomerization for all the variants with 46% BRCA2 ΔDBD, 54% BRCA2 ΔCTD, and 44% BRCA2 ΔDBDΔCTD present as assemblies larger than tetramers (*Figure 4B*, left panel and *Figure 4—figure supplement 1*). Decrease in large oligomers coincides with an increase in the monomer population from <10% in full-length to ~30% in BRCA2 ΔDBDΔCTD

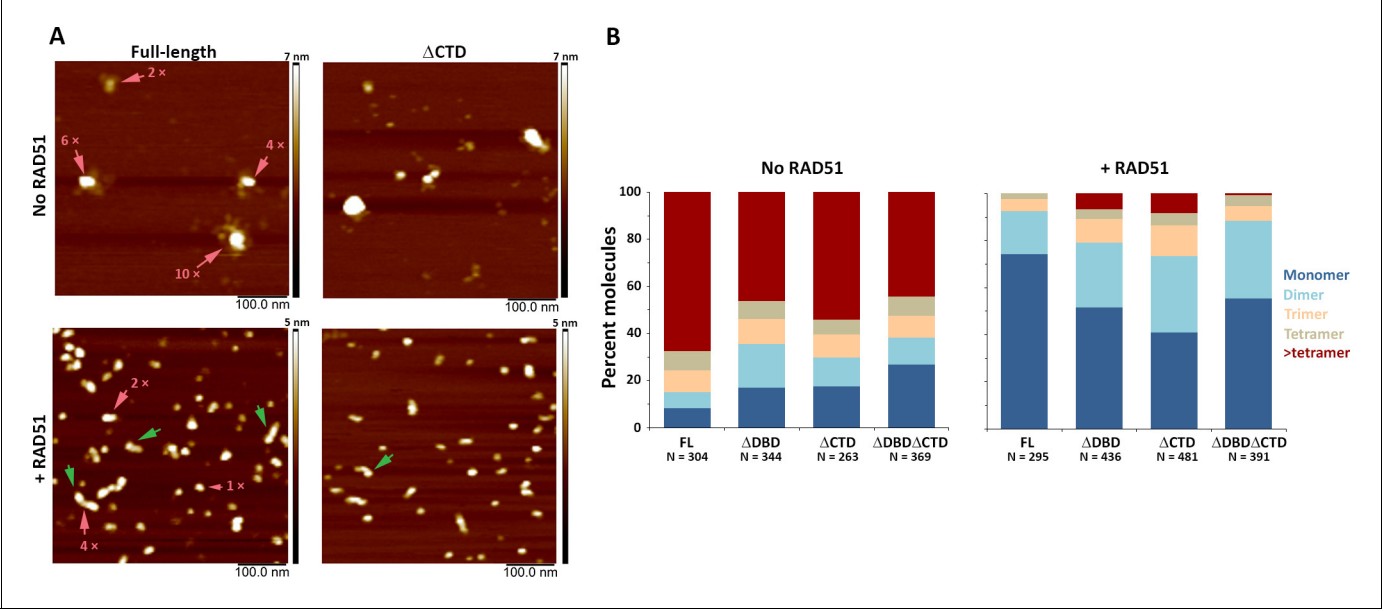

**Figure 4.** C-terminal region of human BRCA2 contributes to the formation of BRCA2-BRCA2 oligomers. (**A**) Representative scanning force microscopy (SFM) height images of full-length and ΔCTD BRCA2 in the presence and absence of RAD51. BRCA2 ΔCTD forms rod-shaped assemblies, like full-length BRCA2, on interaction with RAD51. Rod-like assemblies are indicated by green arrows; pink arrows indicate multimeric assemblies, based on volume analyses. (**B**) Histograms showing oligomeric distribution of full-length BRCA2 and the C-terminal variants in the presence and absence of RAD51. The deletion of C-terminal region leads to lesser oligomeric forms than full-length BRCA2. All the experiments were performed twice with independent protein preparations, imaging, and analyses. The figure is plotted from one of the duplicate data sets. Both data sets can be found in *Figure 4—source data 1*.

The online version of this article includes the following source data and figure supplement(s) for figure 4:

**Source data 1.** Excel files with the source data of the data in *Figure 4B* and the replicate experiment.
**Figure supplement 1.** Representative SFM height images of BRCA2 ΔDBD and BRCA2 ΔDBDΔCTD in the presence and absence of RAD51.
**Figure supplement 2.** Plots showing distribution of full-length BRCA2 and c-terminal deletion constructs in the presence and absence of RAD51.
**Figure supplement 3.** Control scanning force microscopy experiments of RAD51 and ssDNA alone.
**Figure supplement 4.** SDS-PAGE gels visualizing the purified protein preparations as used in this study (*Figure 4* and *Figure 5*).

(*Figure 4B*, left panel), indicating that both DBD and CTD contribute to BRCA2 interactions with itself, at least in the absence of other binding partners.

## DBD and CTD are needed for ssDNA- and RAD51-induced architectural rearrangement of BRCA2

Both RAD51 and ssDNA induce notable changes in BRCA2 architecture. Upon incubation with RAD51, full-length BRCA2 assemblies become largely monomeric (74%) and adopt a more regular compact conformation, with 33% having a rod-like shape (major to minor axis ratio >1.5) (*Figure 4A, B*, *Figure 4—figure supplements 1* and *2*, and *Supplementary file 2*). Purified BRCA2 binds six RAD51 monomers in conditions similar to ours (*Jensen et al., 2010*; *Liu et al., 2010*). Our volume-based monomer designation refers to one BRCA2 plus RAD51, as the theoretical volume of one BRCA2 and one to six RAD51 molecules falls in the range of BRCA2 monomer (see 'Materials and methods' for a detailed description of volume analysis). All deletion variants also become largely monomeric upon interaction with RAD51, but to a lesser extent than the full-length BRCA2 (40–55% for variants vs 74% for full-length). However, all the variants included about one-third of the complexes as dimers: BRCA2 ΔDBD (28%), BRCA2 ΔCTD (32%), and BRCA2 ΔDBDΔCTD (33%), which was more than the full-length BRCA2 (18%) (*Figure 4B,C*, right panel). Only BRCA2 ΔCTD-RAD51 formed rod-shaped assemblies similar to full-length BRCA2 (*Figure 4B* and *Supplementary file 2*). Removing either DBD, CTD, or both reduced BRCA2 oligomerization and, to a lesser extent, reduced RAD51-induced changes in oligomerization and architecture (*Figure 4—figure supplements 1* and *2*).

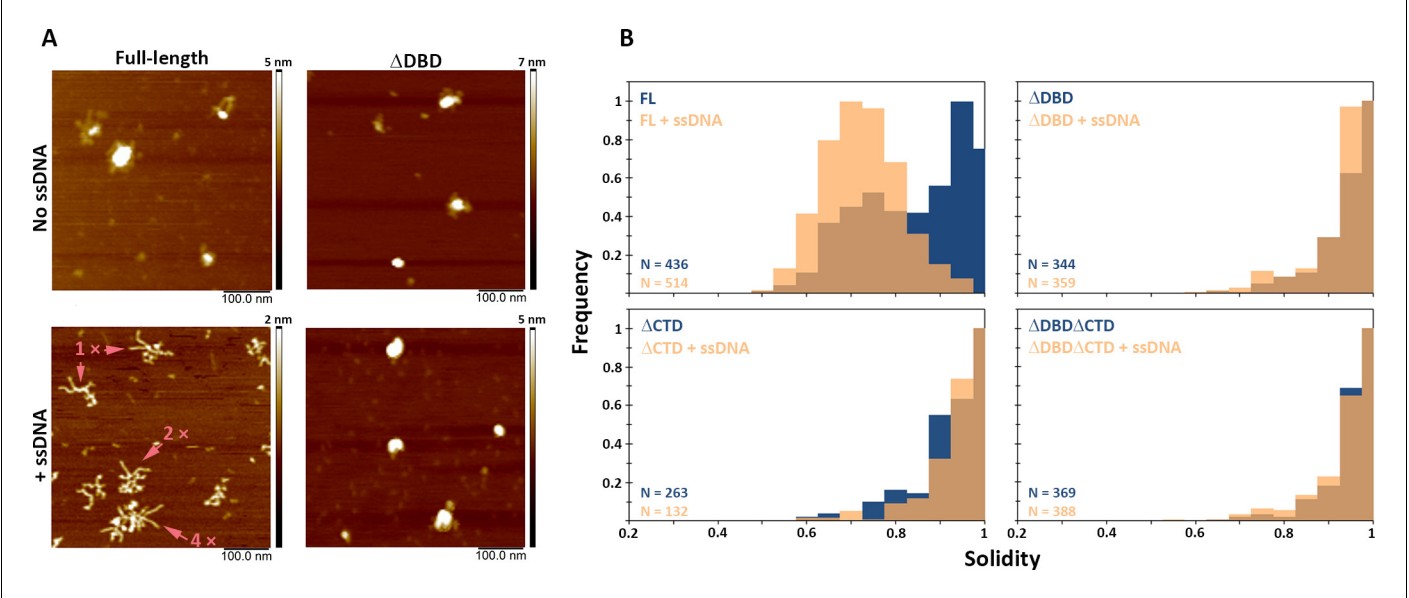

**Figure 5.** C-terminal region of BRCA2 is essential for conformational rearrangement on interaction with ssDNA. (**A**) Representative scanning force microscopy (SFM) height images of full-length BRCA2 and BRCA2 ΔDBD in the presence and absence of single-stranded DNA (ssDNA). Full-length BRCA2 rearranges into extended molecular assemblies on interaction with ssDNA; however, BRCA2 ΔDBD and other C-terminal constructs do not show any conformational change. Pink arrows indicate the oligomeric volume of the particle with respect to the BRCA2 monomer. (**B**) Distribution of full-length BRCA2 and the C-terminal deletion constructs with respect to their oligomerization and solidity. Full-length BRCA2 rearranges to form extended dimers and tetramers on interaction with ssDNA, whereas the deletion constructs do not show any change in their distribution. All the experiments were performed twice with independent protein preparations, imaging, and analyses. The figure is plotted from one of the duplicate data sets. Both data sets can be found in *Figure 5—source data 1*.

The online version of this article includes the following source data and figure supplement(s) for figure 5:

**Source data 1.** Excel files with the source data of the data in *Figure 4B* and the replicate experiment.

**Figure supplement 1.** Representative SFM height images of BRCA2 ΔCTD and BRCA2 ΔDBDΔCTD in the presence and absence of ssDNA.

**Figure supplement 2.** Distribution of full-length BRCA2 and the C-terminal deletion constructs with respect to their oligomerization and solidity.

**Figure supplement 3.** Control experiment showing representative scanning force microscopy (SFM) height images of full-length BRCA2 ± ssDNA in the absence of spermidine, showing that the conformational change observed on interaction with single-stranded DNA (ssDNA) is not an artifact due to presence of spermidine, which is used to facilitate adsorption of DNA on the mica surface for SFM imaging.

While in the presence of RAD51, BRCA2 forms compact structures, ssDNA induces the opposite, characteristic extended forms of BRCA2 become prevalent (*Sánchez et al., 2017*; *Sidhu et al., 2020*; *Figure 5—figure supplement 3*). Full-length BRCA2 is present in oligomeric complexes with extensions, in a bimodal distribution of solidity—a measure of the compactness of the protein complexes—with peaks at 0.75 and 0.95 (*Figure 5A and B*). Incubation with ssDNA shifts the solidity distribution to a single peak at 0.7 (*Figure 5*; FL BRCA2 + ssDNA). The BRCA2 C-terminal deletion variants had fewer extended molecules to begin with, as solidity shows a peak distribution around 0.9 for all the three variants; ΔDBD, ΔCTD, and ΔDBDΔCTD (*Figure 5* and *Supplementary file 2*). In striking contrast to the full-length BRCA2, interaction with ssDNA did not change the distribution of oligomers and shape (solidity) for any of the deletion variants (*Figure 5*, *Figure 5—figure supplement 1*). Both DBD and CTD have to be present for BRCA2 to undergo the conformational change associated with ssDNA interactions. The effect of the BRCA2 DBD and CTD domain deletions on cellular response to DNA damage (*Figures 1–3*) and their effect on the architecture of BRCA2 and its complexes with RAD51 and ssDNA did correlate. The architectural changes we defined here may report on important BRCA2 cellular functions.

## Discussion

Here we investigated the role of DBD and CTD on the diffusive behavior and ligand-induced structural plasticity of BRCA2. We correlated these observations with functional consequences of deleting

these domains, individually and in combination, in living cells. Our panel of isogenic precision-engineered cell lines allowed us to label endogenously expressed BRCA2 directly via the HaloTag. We found that a substantial reduction in DNA-damage resistance, especially in DBD-deficient cells, was accompanied by only subtle changes in dynamics and localization of BRCA2. In contrast, the ability of purified recombinant BRCA2 to undergo structural rearrangements was strongly affected by DBD or CTD deletions (*Table 1*, *Table 2*).

Despite their adjacent location in the C-terminal part of BRCA2 (and frequent simultaneous loss due to human cancer-predisposing mutations), DBD and CTD are functionally distinct. CTD, although much shorter than the DBD, performs several distinct functions: cell-cycle-controlled phosphorylation-dependent stabilization of RAD51 filament in vitro, replication fork protection from excessive nucleolytic processing, and nuclear import. In mouse BRCA2, an additional nuclear localization signal is present at the N-terminus, but in the human protein, there is no such redundancy, which exaggerates the consequence of even short C-terminal truncations, because these produce human BRCA2 that cannot localize to the nucleus (*Sarkisian et al., 2001*; *Spain et al., 1999*). Controlled deletions, including the internal DBD deletion, allowed us to avoid some of the confounding effects complicating previously used mutant or patient cell models.

The DBD is the evolutionarily defining part of BRCA2, conserved from fungi to humans, but its function is less defined than that of other 'younger' BRCA2 regions. Information on DBD function focuses on its interaction with an intrinsically disordered acidic protein DSS1. Our findings reinforce the notion that despite its deep phylogenetic roots, the DBD is not what makes BRCA2 essential for general viability of animal cells. Absence of the DBD leads to significant sensitization to DNA interstrand crosslinks, PARP inhibitor, and radiation, not further exacerbated by additional CTD deletion (*Figure 1*). The role of DSS1 interaction remains puzzling. On the one hand, it is as conserved as the DBD itself, and it was shown to be required for BRCA2 stability and intracellular localization (*Li et al., 2006*); mutations in DSS1 binding phenocopy BRCA2 deficiency, as does DSS1 depletion (*Zhao et al., 2015*). But, on the other hand, in fungi, DSS1 is only required for DNA repair when DBD is present (*Kojic et al., 2005*). Similarly, in human cells, HR could be partially restored in BRCA2-deficient cells by complementation with variants lacking the DBD (*Edwards et al., 2008*; *Siaud et al., 2011*). Our results also show that cells expressing BRCA2 ΔDBD retained ~50% of HR activity (*Figure 1G*). In cells that lack BRCA2 DBD (ΔDBD and ΔDBDΔCTD), we do observe an increased number of RAD51 foci in the untreated condition (p<0.001), which could indicate increased replication-associated DNA damage. This, however, did not affect the growth rate of the cells. We also found that DBD contributes little to the characteristic constrained diffusion we described previously. Binding of the DBD to DNA could explain slow diffusion and frequent immobilization of BRCA2—with a higher frequency after damage induction. But the effect of loss of DBD on diffusive activity was small and comparable to loss of CTD. Therefore, we conclude that the DBD does not have a significant role in this context. In the BRCA2-containing complex, DNA-binding activity could be redundantly supplied by its interactors. For example, several BRCA2-bound RAD51 molecules provide alternative DNA interaction interfaces (*Jensen et al., 2010*; *Reuter et al., 2014*; *Sánchez et al., 2017*). In contrast to the DBD, the CTD is a recent vertebrate addition to BRCA2. In our assays, its deletion resulted in no phenotype (interstrand crosslink survival), intermediate phenotype (radiation and PARPi survival, HR assay, RAD51 focus number), or the same effect as DBD deletion (RAD51 focus intensity). Except for BRCA2 oligomerization (*Figure 4B*), deleting both domains did not result in an additive effect. It is possible that CTD deletion we created encroaches on or disturbs the structure of the DBD, and (some of) the functional

**Table 1.** Summary of results of the in vivo assays in this study.

|  | Ionizing radiation | DNA crosslinks | PARPi | HR | BRCA2 diffusion | Immobilization | RAD51 focus formation |
|---|---|---|---|---|---|---|---|
| Full-length | + | + | + | + | + | + | ++ |
| ΔDBD | - | - | – | - | + | + | + |
| ΔCTD | +/- | + | - | +/- | + | + | + |
| ΔDBDΔCTD | - | - | – | - | + | +/- | + |

**Table 2.** Summary of results of the in vitro assays in this study.

| | Multimerization | Conformational change | |
| --- | --- | --- | --- |
| | | + RAD51 | + ssDNA |
| Full-length | + | +++ | +++ |
| ΔDBD | +/- | ++ | — |
| ΔCTD | +/- | +++ | — |
| ΔDBDΔCTD | +/- | ++ | - |

consequences we attribute to the CTD deletion result from collateral damage to the DBD. The strongest argument against this is the clear separation of functions between the domains in the interstrand crosslink survival assays (*Figure 1E,F*) where CTD deletion has no effect. This finding also suggests that the described fork protection and RAD51 filament stabilization functions of the CTD are not essential for DNA crosslink repair. This is at odds with studies that describe the role of fork protection in crosslink repair and, in particular, with previous findings in ES cells with different CTD-disrupting Brca2 alleles (*Atanassov et al., 2005*; *Donoho et al., 2003*; *Marple et al., 2006*). Details of the genomic engineering strategies could account for the differences: for example, the more widely used Brca2 lex1/lex2 cells are compound heterozygotes, with a larger deletion in one of the alleles (*Morimatsu et al., 1998*). Another unexpected observation was that CTD deletion blocked structural rearrangements of BRCA2 upon interaction with ssDNA as efficiently as the DBD deletion. One possibility is that N-C terminal interactions that contribute to oligomerization of BRCA2 (*Le et al., 2020*) also affect BRCA2 structure and thereby influence the BRCA2-ssDNA interaction. Finally, it is interesting to note that BRCA2 ΔCTD cells in our hands are sensitive to IR and PARP inhibitors but are not sensitive to DNA-crosslinking agents (MMC and cisplatin); this could indicate that these structural rearrangements are more relevant in the context of BRCA2's function in repair in two-ended double-strand breaks than during repair of DNA crosslinks, possibly due to other interacting proteins during crosslink repair.

We observed RAD51 focus formation in cell lines that were, however, deficient in HR at two-ended double-strand breaks. Despite strong sensitization to radio- and chemotherapeutic agents, only careful quantification of the numbers and intensity of RAD51 foci at multiple time points after radiation revealed subtle differences in the deletion variants. The reduced intensity of RAD51 foci in cells lacking DBD and CTD indicates that the repair process is delayed or reduced at some point beyond delivery of RAD51 by BRCA2 to the sites of damage. This suggests that RAD51 foci quantification, although useful to identify more HR-deficient samples than BRCA1/2 mutations, could occasionally return false-positive results for HR function at two-ended breaks. In and of itself, this observation is not surprising as there will be steps important for HR function downstream of RAD51 focus formation. However, in the context of employing the RAD51 focus-formation assay in pre-clinical and clinical settings with the aim to find BRCAness phenotypes, identification of additional HR markers will be of value.

The DBD and CTD of BRCA2 did markedly affect protein architecture and conformational changes in response to binding partners. These domains contributed to oligomerization, when we removed them, and in DBD and CTD deletion variants, the BRCA2 population was less oligomeric. This is in agreement with a recent study where interaction of N- and C-terminal fragments of BRCA2 is indicated to contribute to oligomerization of BRCA2 (*Le et al., 2020*). However, in our study, the oligomeric forms induced by RAD51 binding remain unchanged, which is likely mediated by the interaction with the intact BRC repeats. The characteristic conformational change of irregular compact particles to the extended architecture of full-length BRCA2 in response to ssDNA was severely impaired in all the investigated deletion variants. Together, the inability of the deletion variants to rearrange in vitro in the presence of ssDNA coupled with impaired HR in vivo suggests that DBD and CTD interactions of BRCA2 are important for optimal BRCA2 activity at the sites of damage. A similar regulatory function is reported for other proteins that interact with BRCA2 such as DSS1, which also affects the conformation of BRCA2 (*Le et al., 2020*). A recent study confirmed that regulation of oligomerization of BRCA2 is also relevant in cells, and variants of unknown

significance with mutations in the DBD that reduce the binding of DSS1 show reduced nuclear localization and appear to increase/reduce oligomerization of BRCA2 (*Lee et al., 2021*). Altogether, these and our study suggests that regulation of RAD51 by BRCA2 is affected by conformational rearrangement of BRCA2 and is mediated at different levels by self-interaction of BRCA2 and its interaction partners.

Comparing all the molecular endpoints we analyzed (diffusion, foci, architecture), we conclude that although none correlated perfectly with the functional outcomes (survival and recombination assay), the magnitude of the effect on architectural plasticity was the closest reflection. We are only starting to tease apart the relationship between structural plasticity and cellular function. BRCA2 function may depend not so much on the existence of one structural form or another but on the lifetime of specific conformations affected by its interactors and local chromatin organization, parameters that will need to be quantified.

# Materials and methods

## Key resources table

| Reagent type (species) or resource | Designation | Source or reference | Identifiers | Additional information |
|---|---|---|---|---|
| Gene (mouse) | Brca2 | Genbank | MGI:109337 | |
| Gene (human) | BRCA2 | Genbank | HGNC:1101 | |
| Cell line (mouse) | IB10, subclone of E14 129/Ola | *Hooper et al., 1987* | IB10, mES | |
| Cell line (mouse) | Mouse ES cells *Rad54 -/-* | *Tan et al., 1999* | | |
| Cell line (human) | HEK293T | | | Adapted to suspension culture |
| Antibody | Rabbit polyclonal anti-RAD51 | *van Veelen et al., 2005* | 2307 | IF: 1:10 000 |
| Antibody | Rabbit polyclonal Anti-BRCA2 | Abcam | ab27976 | WB: mouse BRCA2 (1:1000) |
| Antibody | Mouse monoclonal Anti-HaloTag | Promega | G9211 | WB: 1:1000 |
| Antibody | Mouse monoclonal anti-PARP-1 | Enzo | BML-SA250-0050 | WB: 1:1000 |
| Antibody | Anti-rabbit IgG conjugated with CF568 | Biotium/Sigma | Cat# SAB4600310 | IF 1:1000 |
| Antibody | Donkey αRabbit IgG HR Peroxydase | Jackson Imm Res | Cat# 711-035-152 | WB: 1:2000 |
| Antibody | Sheep αMouse IgG HR Peroxydase | Jackson Imm Res | Cat# 515-035-003 | WB: 1:2000 |
| Antibody | Mouse monoclonal anti BRCA2 | Calbiochem | OP95 | WB: full-length, ΔDBD, ΔCTD, ΔDBDΔCTD proteins (1:500) |
| Recombinant DNA reagent | AAV_Actb HR donor plasmid | *Yao et al., 2017*; Addgene | Plasmid #97317 | |
| Recombinant DNA reagent | px459 | *Ran et al., 2013* | | |
| Recombinant DNA reagent | px459 expressing two gRNAs | *Zelensky et al., 2017* | | Modified from *Ran et al., 2013* |
| Recombinant DNA reagent | BRCA2-HaloTag donor plasmid | This paper | | Knock-in construct HaloTag -F2A-neomycin at mouse BRCA2 C-terminus; available on request from corresponding author |

*Continued on next page*

*Continued*

| Reagent type (species) or resource | Designation | Source or reference | Identifiers | Additional information |
|---|---|---|---|---|
| Recombinant DNA reagent | BRCA2 ΔDBD-HaloTag | This paper | | Knock-in construct HaloTag -F2A-neomycin at mouse BRCA2 C-terminus resulting in deletion of DBD; available on request from corresponding author |
| Recombinant DNA reagent | BRCA2 ΔCTD-HaloTag | This paper | | Knock-in construct HaloTag -F2A-neomycin at mouse BRCA2 C-terminus resulting in deletion of CTD; available on request from corresponding author |
| Recombinant DNA reagent | BRCA2 ΔDBDΔCTD-HaloTag | This paper | | Knock-in construct HaloTag -F2A-neomycin at mouse BRCA2 C-terminus resulting in deletion of DBD and CTD; available on request from corresponding author |
| Recombinant DNA reagent | iRFP720-PCNA | This study | pMP37 pGb-iRFP720-I-PCNA | Expression construct flanked by piggyBac inverted terminal repeats; available on request from corresponding author |
| Recombinant DNA reagent | hyPBase | *Yusa et al., 2011* | | Expressing piggy Bac transposase |
| Recombinant DNA reagent | phCMV1-2MBP-TEV-fl BRCA2 | S Kowalczykoski lab | | Expression clone for 293T HEK cells |
| Recombinant DNA reagent | phCMV1-2MBP-TEV-BRCA2 ΔDBD | This study | | Expression clone for 293T HEK cells; available on request from corresponding author |
| Recombinant DNA reagent | phCMV1-2MBP-TEV-BRCA2 ΔCTD | This study | | Expression clone for 293T HEK cells; available on request from corresponding author |
| Recombinant DNA reagent | phCMV1-2MBP-TEV-BRCA2 ΔDBDΔCTD | This study | | Expression clone for 293T HEK cells; available on request from corresponding author |
| DNA oligo | 90 nt ssDNA oligo | IDT | | See sequence in 'Materials and methods' |
| Commercial kit | Q5 site directed mutagenesis | NEB | Cat# E0554S | |
| Commercial assay or kit | MyTaq Red Mix | Bioline | BIO-25043 | |
| Chemical compound, drug | JF549 HaloTag-ligand | *Grimm et al., 2015* | | Gift from L Lavis |
| Chemical compound, drug | JF646 HaloTag-ligand | *Grimm et al., 2015* | | Gift from L Lavis |
| Chemical compound, drug | EdU (5-ethynyl-2'-deoxyuridine) | | Cat# A10044 | |
| Chemical compound, drug | Atto488-azide | ATTO-TEC GmbH | Cat# AD 488–101 | |
| Chemical compound, drug | Atto568 azide | ATTO-TEC GmbH | Cat# AD 594–101 | |
| Chemical compound, drug | MMC (mitomycin C) | Sigma-Aldrich | Cat# M503 | |

*Continued on next page*

*Continued*

| Reagent type (species) or resource | Designation | Source or reference | Identifiers | Additional information |
|---|---|---|---|---|
| Chemical compound, drug | Cisplatin | Sigma-Aldrich | Cat# P4394 | |
| Chemical compound, drug | Olaparib | Selleckchem | Cat# S1060 | |
| Reagent | FreeStyle 293 expression medium | Gibco | Cat# 10319322 | For growth of 293T HEK cells |
| Reagent | Serum-free hybridoma media | Gibco | Cat# 12045084 | For transfection of 293T HEK cells |
| Software, algorithm | DBD tracking | This paper | | Software for analysis of single-moleucle tracking data Available at: https://github.com/maartenpaul/DBD_tracking (copy archived at swh:1:rev:19f3a47289830cf5dc139061a89627b6165da804, *Paul, 2021*) |
| Software, algorithm | DBD foci | This paper | | Scripts for analysis of foci data using CellProfiler Available at: https://github.com/maartenpaul/DBD_foci/ (copy archived at swh:1:rev:157c7953dbed176a65f2c55db7ad48ebfa7f3f5d, *Pau, 2021*) |
| Software, algorithm | SOS Plugin | *Reuter et al., 2014* | | http://smal.ws/wp/software/sosplugin/ |
| Software, algorithm | DL-MSS | *Arts et al., 2019a Arts et al., 2019b* | | https://github.com/ismal/DL-MSS |
| Software, algorithm | CellProfiler | *Carpenter et al., 2006* | | |
| Software, algorithm | Fiji | *Schindelin et al., 2012* | | |
| Software, algorithm | SFMetrics | *Sánchez and Wyman, 2015* | | http://cluster15.erasmusmc.nl/TIRF-SFM-scripts/ |

## Plasmids for cell experiments

Plasmids containing gRNAs and spCas9 were derived from px459 (*Ran et al., 2013*). As described in *Zelensky et al., 2017*, selected gRNA sequences were incorporated into the px459 vector (*Ran et al., 2013*) by digestion of the vector with AflIII and XbaI. The resulting two fragments*Arts et al., 2019b* (vector backbone and restriction fragment) were separately purified from the gel. The resulting restriction fragment was used as the template for two polymerase chain reactions (PCRs) with overhanging primers containing the required gRNA sequence (see *Supplementary file 1*). Using Gibson assembly, the two fragments and the digested vector backbone were assembled and transformed in *Escherichia coli* (DH5 alpha). The correct integration of the gRNA sequence in the isolated plasmid was validated by Sanger sequencing. For incorporation of the two gRNAs into a single plasmid, px459 was modified to contain two U6 promoters and gRNA sequences separated by a short spacer (*Zelensky et al., 2017*).

The donor template for the C-terminal tagging of BRCA2 with a HaloTag was derived from the plasmid that was used to make BRCA2-GFP knock-in cell lines (*Reuter et al., 2014*). This plasmid contains 3' and 5' homology arms (6.6 and 5.4 kb homology) for integration of the construct at the BRCA2 locus. The GFP sequence was removed by restriction digestion and replaced with the Halo-Tag sequence by Gibson assembly. The HaloTag sequence was obtained by PCR from pENTR4-HaloTag (gift from Eric Campeau; Addgene #29644). The donor plasmids for the ΔCTD, ΔDBDΔCTD-HaloTag, contain a 6-kb homology arm upstream of the deletion, while the downstream homology arm was identical to the full-length construct (see *Figure 1—figure supplement 1*). The ΔDBD donor

construct was made by introducing the coding sequence from exon 27 of mouse BRCA2, excluding the stop codon in the ΔDBDΔCTD-HaloTag donor construct.

The PiggyBac iRFP720-PCNA construct was generated using Gibson assembly, by inserting the iRFP720 sequence (*Shcherbakova and Verkhusha, 2013*) and hPCNA sequence (*Essers et al., 2005*), which includes an additional nuclear localization signal sequence in a PiggyBac vector (*Zelensky et al., 2017*) containing a CAG promoter and PGK-puro selection cassette. iRFP720 was obtained by PCR from iRFP720-N1 (gift from Vladislav Verkhusha; Addgene #45461).

## Cell culture

Wild-type (IB10, subclone of E14 129/Ola; *Hooper et al., 1987*) and Rad54-/- mouse ES cells (*Essers et al., 1997*) were cultured on gelatinized plates (0.1% porcine gelatin (Sigma)). The culture media consisted of 50% Dulbecco's Modified Eagle Medium (DMEM) (high-glucose, ultraglutamine; Lonza), 40% Buffalo rat liver cell-conditioned medium, 10% fetal calf serum (FCS) supplemented with non-essential amino acids, 0.1 mM β-mercaptoethanol, pen/strap, and 1000 U/ml leukemia inhibitory factor (mouse). Cell lines were routinely tested (negative) for mycoplasma contamination.

For imaging, cells were seeded in eight-well glass-bottom dishes (Ibidi), which were coated with 25 µg/ml laminin (Roche) for at least 1 hr. About 30,000 cells in 300 µl medium were plated per well the day before the experiment. Cells treated with IR were irradiated in an Xstrahl RS320 X-Ray generator (195.0 kV and 10.0 mA) at the indicated dose.

## Generation of HaloTag knock-in cell lines

15 µg of circular donor plasmid and 15 µg of px459 containing Cas9 and the indicated gRNA(s) were electroporated into $10^7$ IB10 mouse ES cells. About 24 hr after electroporation, cells were put on a selective medium containing 200 µg/ml G418 (Formedium). Medium was refreshed regularly, at least once every second day, and after 8–10 days, colonies were picked into a gelatin-coated 96-well plate. After 2 days, cells in the 96-well plate were split and part of the cells were incubated in lysis buffer (50 mM KCl, 10 mM Tris-HCl, pH 9, 0.1% Triton X-100, 0.15 µg/ml proteinase K) at 50°C for 1 hr. After inactivation of proteinase K at 95°C for 10 min, cell lysates were diluted and 5 µl was used for genotyping PCR using MyTaq DNA polymerase (Bioline) with indicated DNA primers. Selected (homozygous) clones were expanded and knock-ins were validated by western blot as described in *Reuter et al., 2014* on a 5% sodium dodecyl sulphate–polyacrylamide gel electrophoresis (SDS-PAGE) gel using rabbit polyclonal anti-BRCA2 (Abcam, ab27976). Anti-HaloTag (mouse monoclonal; Promega G9211) blot with anti-PARP1 (mouse monoclonal; Enzo BML-SA250-0050) as the loading control (*Figure 1B*) was run on a NuPAGE 3–8% Tris-acetate gel (Invitrogen). From selected clones, genomic DNA was isolated using phenol extraction and additional genotyping PCRs (*Figure 1—figure supplement 1*) were done as described above.

## Clonogenic survivals

For clonogenic survivals, between 100 and 15,000 cells were seeded in gelatin-coated six-well plates. The next day about 16 hr later, cells were treated at the indicated doses with IR or the next day incubated for 2 hr with mitomycin C (Sigma-Aldrich; M503) or for 24 hr with olaparib or cisplatin, after which the cell medium was refreshed. 5–7 days after treatment, the cells were stained with Coomassie Brilliant Blue and manually counted.

## Homologous recombination assay

The AAV_Actb HR donor plasmid used for the Cas9-stimulated HR gene-targeting assay was a gift from Hui Yang (Addgene plasmid #97317) and consisted of 800-bp homology arms targeting the β-actin locus with the P2A-mCherry sequence between the homology arms, as described in *Yao et al., 2017*. The gRNA targeted the same sequence (agtccgcctagaagcacttg) as in the original paper and was cloned into px459 (*Ran et al., 2013*) as described above for the other Cas9/gRNA constructs.

250,000 cells were seeded in 24-well plates and were directly transfected with Lipofectamine 3000, using manufacturer's instructions, using 0.5 µg donor, 0.5 µg px459, and 0.1 µg pGB-TagBFP2. 24 hr after transfection, the medium was refreshed. Cells were measured by flow cytometry (BD LSRFortessa) 4 days post transfection to determine the efficiency of homologous integration of the P2A-mCherry sequence. After gating for single live cells, transfected cells were gated based

on BFP2 expression and, subsequently, the percentage of mCherry-positive cells was determined (*Figure 1—figure supplement 2*). We confirmed, by transfection of the donor plasmid without gRNA and Cas9, that positive cells were not due to background expression of the donor plasmid.

## Immunofluorescence

Cells were grown in eight-well glass-bottom dishes (80826; Ibidi) as described above. When indicated, BRCA2-HaloTag cells were incubated with 250 nM JF549 or JF646 HaloTag ligand. Cells were washed with phosphate-buffered saline (PBS) and fixed in 4% paraformaldehyde (PFA) in PBS for 15–20 min. Cells were washed with 0.1% Triton in PBS and blocked in blocking buffer (PBS with 0.5% bovine serum albumin (BSA) and 1.5 g/l glycine). Primary antibodies were diluted in blocking buffer and incubated with the sample for 2 hr at room temperature. Slides were washed in PBS with 0.1% Triton and subsequently incubated with secondary antibodies in blocking buffer for 1 hr at room temperature. Cells were washed in PBS and DNA was labeled by incubation with 4 ′, 6-diami-dino-2-phenylindole (DAPI) (0.4 µg/ml).

For RAD51 focus quantification on replicating cells specifically, 15 min before fixation, cells were treated with 20 µM EdU in the medium at 37℃. Cells were washed with PBS, pre-extracted for 1 min (300 mM sucrose, 0.5% Triton X-100, 20 mM 4- (2-hydroxyethyl) -1-piperazineethanesulfonic acid (HEPES) KOH (pH 7.9), 50 mM NaCl, 3 mM $MgCl_2$), washed with PBS, and directly fixed in 4% formaldehyde in PBS. For EdU click-chemistry labeling, cells were washed with 3% BSA in PBS, permeabilized with 0.5% Triton in PBS for 20 min. After another wash with 3% BSA, samples were incubated in home-made click-labeling buffer (50 mM Tris, 4 mM $CuSO_4$, 10 mM ascorbic acid, and 60 µM Atto568 azide (ATTO-TEC GmbH)) for 20 min in the dark. For BRCA2 focus quantification, in replicating cells, Atto488 (ATTO-TEC GmbH) was used instead. Subsequently, immunofluorescence was performed as described above and DNA was stained using DAPI.

## Confocal microscopy

Confocal images were acquired at a Zeiss Elyra PS1 system with an additional confocal scan unit coupled to an Argon laser for 488 nm excitation (Alexa 488) and additional 30 mW 405 nm (DAPI), 10 mW 561 nm (CF568, JF549), and 633 nm (Alexa 647, JF646) lasers. A x63 (NA 1.4; Plan Apochromat DIC) objective was used for imaging. At least three positions per condition were selected based on the DAPI signal, and subsequently, automatic multi-position imaging was performed for every position. Fluorescence-based autofocus was used to find the center of the nuclei. A z-stack of 11 slices with 500 nm axial spacing from the center was acquired, while the lateral pixel size was 132*132 nm.

## Foci quantification

BRCA2 and RAD51 foci were automatically quantified using CellProfiler (*Carpenter et al., 2006*). The analysis script can be found at https://github.com/maartenpaul/DBD_foci. In short, from maximum projections of the confocal images, nuclei were segmented using a global threshold (minimum cross-entropy) based on the DAPI signal. Subsequently, within the masked image, based on segmented nuclei, RAD51 foci were identified using global threshold (Robust background) method with two standard deviations above background. The integrated intensity of EdU signal per nucleus was also measured and used to determine the EdU-positive cells. Based on the distribution of the integrated intensity of EdU signal per nucleus, a fixed threshold was set at 500 au; cells above this threshold were defined EdU positive. Also, the integrated intensity per focus for BRCA2 and RAD51 was obtained from CellProfiler. Data were exported as CSV files from CellProfiler. R and Rstudio was used to plot the data (example script can be found at the Github repository mentioned above).

## Live-cell imaging

For tracking experiments, cells were labeled with 5 nM JF549-HaloTag ligand for 15–30 min at 37℃ in mouse ES imaging medium (FluoroBrite DMEM [ThermoFisher], 10% FCS supplemented with non-essential amino acids, 0.1 mM β-mercaptoethanol, pen/strap, and 1000 U/ml leukemia inhibitory factor). Subsequently, cells were incubated twice for 15 min with a fresh imaging medium, while washing the cells once with PBS in between. Microscopy experiments were performed at a Zeiss Elyra PS complemented with a temperature-controlled stage and objective heating (TokaiHit). Samples were kept at 37℃ and 5% $CO_2$ while imaging. For excitation of JF549, a 100 mW 561 nm laser was used.

The samples were illuminated with HiLo illumination by using a x100 1.57 NA Korr αPlan Apochromat (Zeiss) TIRF objective. Andor iXon DU897 was used for detection of the fluorescence signal, and from the chip, a region of 256 by 256 pixels (with an effective pixel size of 100*100 nm) was recorded at 31.25 Hz interval (30 ms integration time plus 2 ms image transfer time). EMCCD gain was set at 300. Per cell, a total of 2000 frames were recorded.

## Single-molecule tracking analysis

Recorded images were converted from LSM format (Zeiss) to tiff in Fiji (*Schindelin et al., 2012*) using the Bioformats plugin and prepared for localization and tracking analysis with the SOS plugin (*Reuter et al., 2014*; http://smal.ws/wp/software/sosplugin/). To track only molecules within the nucleus, for every movie, a mask was manually drawn around the nucleus of the cell. A fixed intensity threshold was used to identify molecules in individual frames. The localized molecules were linked through the nearest neighbor with a maximum displacement of 1.2 µm and a maximum gap size of 1 frame. Tracks had to be at least five frames long to be processed further.

Subsequently, the track data were imported in R for analysis using a home-build script (https://github.com/maartenpaul/DBD_tracking). Tracks were segmented in tracklets using the ML-MSS software described in *Arts et al., 2019a* (https://github.com/ismal/DL-MSS), using a 3-state deep-learning prediction model. Apparent diffusion constants for the tracklets were estimated by determining the slope of the MSD(t) curve from all the tracklets that were at least 10 frames in length.

## Protein expression and purification

Full-length BRCA2 construct in pHCMV1 was a generous gift from S Kowalczykowski. Various variants of BRCA2 (BRCA2 ΔDBD, BRCA2 ΔCTD, and BRCA2 ΔDBDΔCTD) (*Figure 4A*), with two tandem N-terminal maltose-binding protein (MBP) tags, were prepared by Q5 site-directed mutagenesis (NEB) (*Figure 4A*). Purified plasmids were transfected with 10% (v/v) Polyethyleenimine (PEI) transfection solution in 293T HEK cells, adapted for suspension culture, in FreeStyle 293 Expression Medium (Gibco), at approximately $10^6$ cells/ml. Transfection solution was prepared by adding 1 µg/ml purified DNA and 2 µg/ml linear PEI in Serum-Free Hybridoma Media (Gibco) supplemented with 1% FCS. Transfection solution was incubated for 20 min at room temperature and added to 500 ml of HEK cell suspension growing at 37°C, with shaking at 250 rpm. After 48 hr, at a cell count of about $2 \times 10^6$/ml, cells were harvested by centrifugation at 8000 $\times g$, 4°C, for 15 min. The cell pellet was resuspended in 10 ml ice-cold PBS and frozen in liquid nitrogen. Next, cells were lysed in 200 ml lysis buffer (50 mM HEPES (pH 7.5), 250 mM NaCl, 1 % NP-40, 1 mM ATP, 3 mM MgCl₂, 1 mM Pefabloc SC, two tablets of ethylenediaminetetraacetic acid [EDTA]-free protease inhibitor [Roche], and 1 mM dithiothreitol [DTT]) for 15 min at 4°C with shaking. The lysate was centrifuged at 10,000 $\times g$, 4°C, for 15 min. The supernatant was incubated O/N with 10 ml amylose resin pre-equilibrated in wash buffer (50 mM HEPES (pH 7.5), 250 mM NaCl, 0.5 mM EDTA, 1 mM DTT). Next day, the beads were washed three times with wash buffer by centrifugation at 2000 $\times g$ at 4°C for 5 min and aspiration of the supernatant. The washed resin was incubated with elution buffer (50 mM maltose, 50 mM HEPES [pH 8.2], 250 mM NaCl, 0.5 mM EDTA, 10% glycerol, 1 mM DTT, 1 mM Pefabloc SC) for 15 min at 4°C on a rolling platform. The eluate was collected by passing the slurry through a disposable BioRad column at 4°C. The eluate was loaded on a 1 ml HiTrap-Q column from GE using Q low buffer (50 mM HEPES [pH 8.2], 250 mM NaCl, 0.5 mM EDTA, 10% glycerol, 1 mM DTT, 1 mM phenylmethylsulfonyl fluoride [PMSF]) and eluted with Q high buffer (50 mM HEPES [pH 8.2], 1 M NaCl, 0.5 mM EDTA, 10% glycerol, 1 mM DTT, 1 mM PMSF). Peak elution fractions were checked by western blot using mouse monoclonal anti-BRCA2 (OP95-Calbiochem) as the primary antibody (1:500) and sheep anti-mouse HRP (1:2000) (Jackson ImmunoResearch) as the secondary antibody. Fractions with proteins were aliquoted into single-use aliquots by snap freezing in liquid nitrogen and stored at −80°C. Purity and yield of all the protein preparations was checked by 8% SDS-PAGE analysis; the purified fractions were electrophoresed on the gel and stained with silver stain and Coomassie brilliant blue R-250 (*Figure 4—figure supplement 4*).

Untagged human RAD51 was expressed and purified as described by *Modesti et al., 2007*.

## SFM sample preparation, imaging, and analyses

For BRCA2-RAD51 reactions, aliquots of BRCA2 stored at −80°C were thawed and diluted fourfold in 10 mM HEPES, pH 8.0, buffer to subsequently prepare a reaction of 2.5 nM BRCA2 construct in 22 mM HEPES, pH 8.2, 112 mM NaCl, 0.125 mM EDTA, 2.5% glycerol, and 0.25 mM DTT. Samples were incubated at 37°C in the absence or presence of 250 nM RAD51 for 30 min without shaking.

For BRCA2-ssDNA reactions, after dilution as mentioned above, the protein was incubated at 37°C for 30 min with a linear 90-nt ssDNA oligo (3.4 µM in nt) (5′-AF647/AATTCTCATTTTACTTACCG-GACGCTATTAGCAGTGGCAGATTGTACTGAGAGTGCACCATATGCGGTGTGAAATACCGCA-CAGATGCGT-3′). After incubation, 50 µM spermidine was added to the sample.

Samples for SFM imaging were prepared by depositing 20 µl of reaction volume on a freshly cleaved mica (Muscovite mica, V5 quality, EMS) for 2 min, followed by a 2 ml wash using 18 MΩ water and drying in filtered (0.22 µm) air. SFM images were obtained with a Nanoscope IV (Bruker), using tapping mode in air with a silicon probe, NHC-W, with a tip radius <10 nm and a resonance frequency range of 310–372 kHz (Nanosensor; Veeco Instruments, Europe). All images were acquired with a scan size of $2 \times 2$ µm at $512 \times 512$ pixels per image at 0.5 Hz. Images were processed using Nanoscope analysis (Bruker) for background flattening. Quantitative analysis of the images was performed as described using SFMetrics software (*Sánchez et al., 2017*; *Sánchez and Wyman, 2015*; *Sidhu et al., 2020*). In volumetric analyses, a comparison of the oligomeric volume of the different regions with RAD51 ($56 \text{ nm}^3$) showed that the monomer volume of RAD51 is much lower than the threshold volume and, thus, free RAD51 is removed from analysis (*Figure 4—figure supplement 1*).

The conformation of the molecules was quantified by the parameters of solidity. Solidity measures the irregular shape of the selected molecule by using the ratio of the area of the selected molecule to the area of a convex hull, which completely encloses the molecule. Solidity is presented in a scale of 1–0, where a value of ~1 signifies a globular molecule while a value ~0 represents a highly irregular molecular shape.

## Acknowledgements

We thank the Optical Imaging Centre for use and technical assistance with the optical microscopes; Ihor Smal (Erasmus MC) for assistance in single-molecule tracking analysis; Luke Lavis (HHMI Janelia) for providing HaloTag ligands; Niklas Bachmann for assistance in making the BRCA2-Halo ΔCTD cell line. We thank Joyce Lebbink (Erasmus MC) and Nick van der Zon (Erasmus MC) for critically reading the manuscript. We acknowledge the Josephine Nefkens Cancer Program for infrastructural support.

## Additional information

### Funding

| Funder | Grant reference number | Author |
|---|---|---|
| Nederlandse Organisatie voor Wetenschappelijk Onderzoek | | Maarten W Paul |
| KWF Kankerbestrijding | 10436 | Arshdeep Sidhu |
| KWF Kankerbestrijding | 11143 | Yongxin Liang |
| Cancer Genomics Centre | | Alex N Zelensky |
| Convergence Health & Technology | CHT16 | Maarten W Paul |

The funders had no role in study design, data collection and interpretation, or the decision to submit the work for publication.

### Author contributions

Maarten W Paul, Conceptualization, Resources, Data curation, Software, Formal analysis, Validation, Investigation, Visualization, Methodology, Writing - original draft, Writing - review and editing; Arshdeep Sidhu, Conceptualization, Resources, Data curation, Formal analysis, Validation, Investigation,

Visualization, Methodology, Writing - original draft, Writing - review and editing; Yongxin Liang, Formal analysis, Investigation, Writing - review and editing; Sarah E van Rossum-Fikkert, Hanny Odijk, Resources, Investigation, Writing - review and editing; Alex N Zelensky, Conceptualization, Resources, Data curation, Formal analysis, Supervision, Investigation, Visualization, Methodology, Writing - review and editing; Roland Kanaar, Conceptualization, Resources, Supervision, Funding acquisition, Methodology, Writing - review and editing; Claire Wyman, Conceptualization, Resources, Supervision, Funding acquisition, Validation, Methodology, Writing - original draft, Project administration, Writing - review and editing

### Author ORCIDs
Maarten W Paul ⓘ https://orcid.org/0000-0002-7990-6010
Arshdeep Sidhu ⓘ http://orcid.org/0000-0002-2851-1019
Roland Kanaar ⓘ http://orcid.org/0000-0001-9364-8727
Claire Wyman ⓘ https://orcid.org/0000-0003-2549-6893

### Decision letter and Author response
Decision letter https://doi.org/10.7554/eLife.67926.sa1
Author response https://doi.org/10.7554/eLife.67926.sa2

## Additional files

### Supplementary files
• Supplementary file 1. Tables with gRNA sequences used for generating the BRCA2-Halo knock-in cell lines and primers used for genotyping the cell lines.

• Supplementary file 2. Summary of average solidity of BRCA2 variants in the absence and presence of ssDNA (90-mer) oligomer.

• Supplementary file 3. Percent rod-like assemblies in BRCA2-RAD51 interaction in various constructs.

• Transparent reporting form

### Data availability
All data generated or analysed during this study are included in the manuscript and supporting files. Source data files have been provided for Figures 1–5.

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
