## [Decision Letter]

**Acceptance summary:**

This work is of interest to readers in the field of genome stability, DNA repair and associated human diseases. The manuscript describes systematic analyses of the crucial DNA repair mediator BRCA2 and its variants lacking the DNA binding domain or RAD51 interacting C-terminal domain, and the conclusions present a conceptual advance as to how BRCA2 promotes DNA repair. The work is a technical tour de force that includes evaluation of the DNA damage response, gene targeting and single particle tracking in mouse embryonic stem cells, as well as biophysical analyses of the human counterparts.

**Decision letter after peer review:**

Thank you for submitting your article "Role of BRCA2 DNA-binding and C-terminal domain on its mobility and conformation in DNA repair" for consideration by *eLife*. Your article has been reviewed by 3 peer reviewers, and the evaluation has been overseen by Maria Spies as a Reviewing Editor and Jessica Tyler as the Senior Editor. The following individuals involved in review of your submission have agreed to reveal their identity: Fumiko Esashi (Reviewer #1); Sarah R Hengel (Reviewer #2).

Essential Revisions:

The reviewers agree that this is an important, technically challenging and generally expertly executed study. As you will see from the individual reviewers' comments, the following revisions will be required to gain full confidence in the reported claims:

1. Figures 1 and 2 need improvements with biological replicates and statistical tests. The data presented in these figures are promising, although would benefit from three biological replicates.

2. After extensive discussion, all three reviewers had concerns with interpretation of the RAD51 foci data. Figure 2 shows that RAD51 foci formation kinetics is slower in all BRCA2 truncation variants compared to full-length BRCA2 expressing cells. The reviewers felt that the authors somewhat over-interpreted their observations that RAD51 foci are eventually formed in all variants. The observed alteration in the number and intensity of foci at the early time points should be discussed.

3. SDS gels of purified constructs should also be included.

4. There seems to be a discrepancy between the impact of structural changes in CTD truncated BRCA2 (shown in Figures4 and 5) and observed cellular phenotypes (Figure 1. MMC and Cisplatin resistance). The authors should elaborate their discussion on this.

*Reviewer #1:*

The breast cancer protein 2, BRCA2, is best known for its roles in DNA repair by homologous recombination (HR) and in protecting stalled replication forks. In these processes, BRCA2 is thought to play a primary role in delivering RAD51 to affected ssDNA-containing sites. However, BRCA2 is a large protein of ~400 kDa, mostly composed of disordered structure, and how it acts during HR repair remains not fully understood.

This work tackles this fundamental question and aims to uncover essential functions of the C-terminal region of BRCA2, composed of the ssDNA binding domain (DBD) and RAD51 binding C-terminal domain (CTD). Using mouse embryonic stem (mES) cells in which BRCA2 DBD and/or CTD deletion variants are endogenously expressed as HeloTag fusions, the authors systematically analysed (1) cellular survival upon genotoxic treatments and HR competency, (2) nuclear localisation and (3) diffusion dynamics. The work was further extended to the structural analyses of purified human BRCA2 with analogous deletions, assessing the impact of RAD51 or ssDNA for their conformational changes.

The authors show that, while DBD and CTD are both important for normal cellular survival upon DSB-inducing IR and for HR activity, these deletion variants are capable of forming RAD51 or BRCA2 foci and mobility changes following IR, comparably to full-length BRCA2. Conversely, they found the clear impact of deletion of DBD or CTD in their oligomeric states and structural plasticity.

Together, the authors conclude that the cellular survivals upon IR and HR competency are best reflected by the BRCA2 structure plasticity, rather than RAD51/BRCA2 foci formation or mobility. Accordingly, the authors propose that BRCA2's role in promoting HR is not simply delivering RAD51 to DNA damage sites, but requires its conformational changes. This also raises a caution to the widely used readouts, such as RAD51 foci formation, to infer the functionality of BRCA2. Overall, I feel that their conclusion is justified by the results presented in this manuscript.

The major strength of this work lies in their comprehensive analyses of BRCA2 variants using a wide range of state-of-the-art in vivo and in vitro techniques, allowing straightforward comparison of their impact on cellular function, molecular behaviour and structural changes. The limitations of this study, although minor for the conclusion drawn by this study, are (1) CTD deletion generally confers modest cellular phenotypes compare to DBD deletion and is fully resistant to MMC and cisplatin. It remains unknown why CTD deletion elicits less impact despite its strong impairments in ligand-induced conformational changes; and (2) the molecular behaviours of BRCA2 in mouse ES cells might not be directly translated to these in human somatic cells.

My specific comments on each experimental data are outlined below:

(1) Survival assays of respective mES cell lines show that CTD is important for normal resistance to IR and olaparib, but not for MMC or cisplatin, while DBD is important for all aforementioned treatments. Their analysis of HR competency, inferred by Cas9-induced gene targeting efficiency, revealed that the deletion of DBD, and of CTD to a lesser extent, impact on efficient integration of the reporter, concluding that these domains are important for HR repair of two-ended DSB. These results are robust and convincing.

(2) They then moved onto the analyses of the IR-induced RAD51 and BRCA2 foci formation. Surprisingly, they found that the deletion of DBD or CTD did not drastically affect foci formation, albeit slightly less efficient compared to full-length BRCA2. While the results and trends look promising, the number of samples analysed is somewhat limited (i.e., two or three technical replicates, rather than biological replicates) and the statistic tests have not been conducted.

(3) HeloTag also allowed them to assess the mobility of these BRCA2 variants in mES cells, using single-particle tracking (SPT). Focusing on S-phase cells, they show that the increase of the immobile fraction of BRCA2, detectable at 2-4 hours upon ionising radiation, is not severely affected by the deletion of DBD or CTD. The conclusion was drawn from the datasets from two independent experiments of at least 15 cells and ~10,000 tracks per condition, which, in my opinion, is respectful.

(4) Equivalent human BRCA2 deletion variants were purified from human HEK293 cells and subjected to scanning force microscopy (SFM) imaging. This analysis revealed that, while full-length BRCA2 commonly forms large oligomers of more than four molecules (70%), all the truncation variants showed somewhat reduced capacity to form tetramers or larger oligomers (i.e. ~44-54%). Upon RAD51 incubation, the majority of full-length BRCA2 (74%) became monomeric, while the C-terminal deletion appeared to respond less, with 40-55% becoming monomers and 30% remaining as dimers. The addition of ssDNA made full-length BRCA2 structure extended but elicited no structural impact on the truncated variants. These conclusions were drawn from the analysis of ~260-500 particles per sample, and look to me, credible. It would nevertheless be good to see the quality of purified BRCA2 variants by silver staining or mass-spectrometry to eliminate potential complications associated with other co-purified factors.

Line 61 “remarkable rearrangement by RAD51, ssDNA and ssDNA” – duplication of ssDNA

Line 79 'phosphorylation-dependent, RAD51 interaction domain' – RAD51 interaction with CTD is 'blocked' by the phosphorylation at S3291. The sentence might be misread as 'CTD phosphorylation promotes RAD51 interaction'. Make this clear at least for the first time when it is refereed.

Figure 1 – Colour codes do not always match. For example, in panels A and G, δ-DBD is shown in blue, but in panels C-F, it is shown in purple. It will be easier to read the results if they are shown in consistent colour codes.

Figure 2 – This figure was hardest for me to understand due to three main issues. Firstly, while the manuscript discusses the kinetics of foci formation at any given time point, the results are clustered according to the type of samples (i.e. full-length, dDBD, dCTD and dDBD/dCTD), rather than each time points. Clustering treatment or time point (i.e. with and without IR, or hours post-IR) would make it easier to understand. Secondly, statistic tests had not been conducted, hence it is difficult to assess the conclusions are supported by their results. In fact, the deletion mutants seem to show significant difference from full-length BRCA2 at 2 hour after IR. Finally, as they are assessing the action of BRCA2 in the context of RAD51 recruitment, they could include the quantitative assessment of BRCA2-RAD51 co-localising foci. This can be done by ImageJ co-localisation plugin.

Figures 4 and 5 – These results are well presented in my opinion. The manuscript, however, goes back and forth to explain these datasets, i.e., first explains BRCA2 oligomer formation (page 15), followed by solidity of BRCA2 affected by ssDNA (page 16), then back to BRCA2 oligomer status affected by RAD51 (page 17). It will be more straightforward if the data shown in Figure 4 are explained fully, before moving to the solidity assessment shown in Figure 5.

The beauty of this study is its consistency in the samples analysed. It will be good to include a table, summering the phenotypes and molecular properties of each BRCA2 variants.

*Reviewer #2:*

In the manuscript by Paul W. Maarten and Sidhu A. et al., the authors surveyed the importance of the DNA binding domain (DBD) and C-terminal domain (CTD) of BRCA2 in response to DNA damaging agents in cells, and the conformations adopted by recombinant constructs. The characterization of these domains are paramount in understanding basic BRCA2 function for novel future exploitation in cancer therapeutics. While the DBD and CTD domain have notable functions in DNA binding, nuclear localization upon DSS1 binding, RPA exchange, and replication fork protection, their role in response to damage and conformational modulation had been unexamined. Studying BRCA2 domain deletion in human cell lines is difficult as human BRCA2 contains a NLS in the C-terminus of the protein. The authors exploit the fact that the murine BRCA2 that has an additional N-terminal nuclear localization sequence to overcome lethality and the study of deletion mutants in human cell lines. Cell survival assays show the DNA binding domain of BRCA2 is most important for cell survival when treated with DNA damaging agents IR, Olaparib, MMC, and Cisplatin. The authors also show this system is functional as they observe the DBD domain is the most important for gene targeting assays that are repaired by homologous recombination. By assessing various DNA damaging agents, the authors highlight the multiple roles of BRCA2 in varying DNA repair processes from DSB repair, BIR, crosslink repair, etc. Interestingly, the C-terminus of BRCA2 does not appear to play a role in to cells when treated with MMC or Cisplatin but plays an important role in mediating self-organization. The authors describe that both the DBD and CTD domains of BRCA2 are important for RAD51 foci formation following IR. Assessing BRCA2 single-particle tracking in live cells, the authors show that the deletion of the DBD and the CTD domain leads to an increased immobile fraction following IR treatment. Using biophysical single molecule analysis, the authors analyzed recombinant BRCA2 DBD , CTD, and double mutants in the presence of ssDNA and interacting protein RAD51. The authors determined these domains are important for BRCA2 self-interactions and BRCA2 conformational rearrangements in the presence of ssDNA supporting in vivo analysis. Biophysical analysis show that the DBD and CTD are important for BRCA2 conformational dynamics that are observed with binding protein RAD51 or DNA substrates.

Strengths:

– These studies exploit a murine cellular system to overcome cellular lethality observed in BRCA2 depletion in human cell lines, which allows them to study the mouse BRCA2 protein and associated domain deletions.

– The authors also utilize bright photostable fluorophore's called JF646 Halo Tag ligand to study BRCA2, the deletion mutants, and RAD51 using live cell imaging. This is a great technical advancement in observing BRCA2 function in vivo.

– The in vivo and in vitro studies both support important roles of the DBD and CTD domain in BRCA2 dynamics.

Weaknesses:

– The importance of the in vivo work with these domains and the findings presented is confounded by a lack of biological replicates and clear presentation of statistical analysis within figures in the manuscript.

– As both domains are important for response to DNA damaging agents (IR, Olaparib, MMC, and Cisplatin) if a function specification could be made to the deletion mutations this would be most valuable to the field. Assaying molecules with varying substrates (Ex-forked substrates, crosslinked substrates, ssDNA substrates containing DNA lesions) or other protein players (DSS1) may aid in teasing out these roles.

– The discussion focuses on DSS1 and the DBD domain, yet the paper lacks any experimental analysis of BRCA2-DSS1. A biophysical analysis with recombinant protein DSS1 may greatly enhance the impact of this work on the field.

– It is unclear if the larger BRCA2 assemblies or the deletion mutants in the manuscript form via an oligomerization mechanisms or a phase separated mechanism. Speculation from authors would be valuable.

1) Figure 1 and 2: It is a standard in the field that cellular experiments presented should have at least three biological replicates for chlonogenic survival assays. Please perform the biological and technical replicates for data presented in Figure 1 and Figure 2.

a. Figure 1:

i. More than two concentrations of (Olaparib) should be assayed in chlonogenic survival to understand the full curve of cellular survival.

ii. Presentation of the statistical analysis directly in the Figure C-F would be helpful in ascertaining importance with a description in the manuscript.

iii. Displaying the chlonogenic data in a non-logarithmic form would be helpful to viewers for comparing the IC50 of survival.

iv. The description of how cellular sensitivities to IR, Olaparib, MMC, and cisplatin relate to repair outcome is lacking. Please describe in more detail how each agent causes damage and the potential outcomes and DNA repair pathways that would be utilized for repair.

b. Figure 2: While the DBD and CTD are not essential for BRCA2 and RAD51 foci formation the data presented in Figure 2E suggest that the DBD plays the most important role. The kinetics of RAD51 foci formation/behavior of the δ DBD shows a different behavior compared to full length BRCA2 and δ CTD.

i. It is unclear if this phenomena is different and could benefit from increased biological replicates and display of error. Statistical analysis for Figures 2B,D,C, and F would be helpful in ascertaining importance.

ii. Moreover, how the authors hypothesize this domain plays a role in RAD51 foci turnover would be helpful as this domain does not directly interact with RAD51 protein.

c. A more mechanistic functional role of these domains can be obtained by single-molecule analysis of BRCA2 and the deletion constructs with varying substrates and BRCA2 binding partners.

i. It is unclear why the CTD is not sensitized to MMC or Cisplatin but previous work in the field has shown a role for this domain in fork protection and crosslink repair (Atanassov et al., etc) as the authors describe in the discussion.

ii. These discrepancies and a mechanism may be discerned by evaluating single-molecule analysis with DSS1 (known DBD interaction partner) and fork substrates with and without crosslinks. Other protein binding partners like DSS1 and p53 etc. may be important for dissecting out function of the studied domains. These revisions should be easily completed by the talented authors within 6 months.

1. For p53: Rajagopalan S. etg al., PNAS 2010 PMID: 20421506

*Reviewer #3:*

The biochemical and genetic characterization of BRCA2 has been an ongoing challenge in the DNA repair field as the protein is large, prone to degradation, and expressed at low levels in most cell types. While certain features of BRCA2 have been described previously including its ability to bind and load RAD51 onto resected DNA substrates, much remains to be discovered. In this study, the authors combine genetic studies in mouse ES cells with biochemical analysis to examine the spatial dynamics and molecular architecture of BRCA2. Notably, they utilize an innovative approach coupling endogenous tagging of mouse BRCA2 with a HALO tag to monitor BRCA2 movement within live cells by single particle tracking.

I applaud the authors for achieving a highly technical approach to epitope tagging both endogenous BRCA2 alleles in mouse ES cells and combining this strategy with a HALO tag providing additional utility for a variety of cell biological experiments. By analyzing the endogenous alleles, the authors' system provides physiological levels of protein expression as transcription will be driven by the endogenous promoter thus preserving stoichiometric protein interactions within the cell and avoiding artifacts caused by overexpression.

The authors determine the influence of the DNA binding domain (DBD) and c-terminal binding (CTD) on the dynamic activities of BRCA2. They begin by exposing cells containing 3 different deletion mutants ∆DBD, ∆CTD, and the double mutant ∆DBD∆CTD to four different types of DNA damage (IR, PARPi, MMC, and cisplatin). Notably, ∆DBD displays significant impairment in survival in response to all 4 types of DNA damage. The ∆CTD, in contrast, demonstrates less sensitivity to IR and Olaparib, however, complements as well as WT BRCA2 in response to crosslinking agents MMC and cisplatin. My only criticism in this aspect of the work is that it would have been informative to include a truncated BRCA2 (mimic of a patient pathogenic mutation) or null allele to compare to the survival of the ∆DBD and ∆CTD mutants. I realize that these alleles may be inviable but the authors should clearly state if that was indeed the case.

The authors then go on to demonstrate that the ∆DBD and ∆CTD mutants are recruited to sites of IR damage in a similar manner to WT BRCA2 based on number and intensity of foci. I think it would be informative if the authors provided statistical significance for the graphs depicting the quantitation of foci number and intensity as there do appear to be differences between the mutants and the WT protein. There appears to be a delay in the kinetics of recruitment, especially at the 2 hr timepoint, for the mutants compared to WT BRCA2, which could indicate a defect in the recognition of the DNA damage. Only at the 2 hr timepoint following IR are there less RAD51 foci, and of a lesser intensity, in the three deletion mutants compared to WT BRCA2. Another possibility is the results could be interpreted as a defect in RAD51 loading and/or stabilization of the nucleoprotein filament. While immunofluorescence imaging of DNA repair foci have become common practice to measure protein recruitment to damage, it is impossible to know exactly what is happening in these foci with any granularity.

Next, the authors measure BRCA2 movement in the mouse ES cells taking advantage of the HALO tag to track single particles. While technically and visually alluring, it is difficult to extract mechanistic insight from the results. DNA damage induces changes in diffusion leading to BRCA2 molecules with restricted mobility; the authors demonstrated this phenomenon in a prior publication. The deletion mutants appear to have little effect upon BRCA2 mobility.

Finally, the authors utilize scanning force microscopy to analyze binding of the purified human BRCA2 proteins to RAD51 and ssDNA. In the absence of RAD51/ssDNA binding, there is a notable shift in the deletion mutants from oligomeric forms to monomeric compared to full length WT BRCA2. Upon binding to RAD51, there is a dramatic change from multimeric to monomeric forms for the WT BRCA2 (~7% to 74%) with a slight suppression of these changes shown for the deletion mutants. While WT BRCA2 forms extended molecular assemblies upon binding ssDNA, not surprisingly, deletion of the DBD or CTD fail to demonstrate any significant changes in physical architecture. In both situations, the mutant proteins respond to RAD51 and ssDNA in a dampened manner likely due to altered or loss of binding. While the architectural effects of RAD51 and ssDNA binding to BRCA2 are measurable by SFM, it is difficult to reconcile these changes in shape and oligomerization to defects in response to DNA damage and at which specific steps in homologous recombination these physical forms would impact.

Strengths:

1. Generation of mouse ES cells with both endogenous alleles of BRCA2 containing the deletion mutations in addition to a HALO tag is an incredible technical breakthrough and will be a highly valuable reagent for genetic and cell biological studies of mouse BRCA2.

2. The deletion mutants ablating either the DBD or the CTD, or both, is a great genetic approach to understanding the role of these key domains in BRCA2. The response of these mutants (versus WT BRCA2 as a benchmark) to various DNA damage (IR, PARPI, MMC, cisplatin) provides interesting information delineating the roles of these two important domains in BRCA2. For example, the ∆CTD mutant is significantly sensitive to IR and Olaparib, yet complements as well as WT BRCA2 in response to the crosslinking agents MMC and cisplatin.

3. The BRCA2 protein is notoriously difficult to purify and yet the authors succeeded in purifying 4 different forms of the protein for biophysical analysis. While it is difficult to interpret the various forms of BRCA2 by SFM, there are clear differences in the architecture between WT and the three c-terminal mutants. These differences are highlighted upon binding to RAD51 or ssDNA.

Weaknesses:

1. While the separation-of-function result for the CTD deletion in response to crosslinking agents MMC and cisplatin is a novel and compelling result, it would have been informative to compare the survival results and gene targeting assay using a BRCA2 null or mimic of patient mutation (truncating mutation) to see how these 3 mutants stack up against a completely non-functioning BRCA2 allele. Likely, the BRCA2 null alleles are inviable but perhaps a conditional system or truncating allele similar to a patient germline mutation would give a window into response compared to the DBD and CTD deletion mutants.

2. It's not clear in the manuscript what new information we are learning about the mechanisms of BRCA2 in the single particle tracking (SPT) data. The differences in mobility between the mutants and WT BRCA2 seem minimal, but more importantly, it is not immediately clear how these data help us understand the normal cellular functions of BRCA2. No doubt, the technology and innovation to track single particle proteins in the nuclei of cells is impressive, but the authors should clearly explain how we can gain mechanistic insight from the SPT data that is presented in this manuscript.

General Comments:

It is unclear how missing the c-terminal domain (CTD) or the DNA binding domain (DBD) of BRCA2 can be interpreted as having "roles beyond delivering strand exchange protein RAD51" unless a complete biochemical workup of the deletion mutants was performed to detect any alterations in DNA binding, stimulation of RAD51 dependent strand exchange, etc… While interesting and certainly an impressive technical feat, foci imaging and single particle tracking do not provide much information on mechanism (i.e. whether BRCA2 is binding DNA and loading/nucleating RAD51).

The interpretations in the discussion are not overstated, however, I somewhat disagree with the notion that the data, as presented, clarifies the role of BRCA2 beyond its canonical functions of RAD51 loading and nucleation on resected DNA substrates. I would have liked if the authors discussed the idea that it is surprising that mouse ES cells can tolerate complete loss of the DBD, CTD, and loss of both together. Questions that should be addressed in include some of the following: Are proliferation rates compromised compared to WT cells? Are they experiencing replication stress in the absence of any exogenous damage? Further, is there something unique about mouse ES cells that may differentiate BRCA2 behavior that would be expected in somatic human cells?

It is interesting to note that many years ago Ashworth and Taniguchi published back-to-back papers in Nature (2008) describing BRCA2 reversion alleles from in vitro screens of BRCA2 mutant cells selected in cisplatin or PARPi such that some of these reversions resulted in huge deletions of the entire DBD of BRCA2, and yet, they promoted resistance to PARPi. In this context, I would much appreciate if the authors commented on their findings that their constructed DBD deletion is not resistant to PARPi and if they offered some speculation as to why the reversions in those previous studies were.

1. Are the BRCA2 mouse alleles missing the c-terminus (∆CTD) properly localized to the nucleus? The authors do mention that mouse BRCA2 contains an NLS on the N-terminus of the protein but did they check? The human BRCA2 ∆CTD protein would be expected in the cytoplasm as NLS are located at the extreme c-terminus of the protein.

2. In the figure 1A schematic, can the HALO tag graphic be placed on the c-terminus of the three deletion mutants? It might be confusing left only on the full length BRCA2 depiction.

3. Do the three purified human BRCA2 proteins bind DNA substrates in an EMSA experiment? It would be informative to know if binding to certain substrates is diminished or ablated, for example, on ssDNA vs dsDNA vs 3' and 5' tails.

4. A similar question is if the purified mutants are compromised for binding RAD51, and more specifically, binding and stabilizing the RAD51 filament as previously demonstrated for the CTD (Esashi et al. 2005 and 2007, Davies et al. 2007)?

5. In the first paragraph of page 5 (lines 74-86), it would be helpful if the authors discussed briefly the separation-of-function of the BRC repeats in BRCA2 (Carreira et al. 2011, Chatterjee et al. 2016) especially in the context that BRC5-8 seems to operate in a mechanistic manner similar to the CTD in stabilization of the RAD51 filament on ssDNA.

6. Figure 1C-E: please increase size of data labels (boxes, circles, triangles) and lines in the survival curve graphs to make it easier for readers to visualize the results.

7. In the survival assays in Figure 1 C-E, why didn't the authors include a classical BRCA2 pathogenic mutation (or null) to use as a benchmark for comparison to ∆DBD and ∆CTD? Inviable? It would be a very informative comparison if possible.

8. Can the authors comment on the 2 bands in Figure 1B western confirming expression of the HALO tagged mutant alleles in mouse ES cells, is one BRCA2 and the other a degradation product, or is one of the alleles truncated on the N-terminus? Were both alleles completely sequenced in the cells used for the studies?

9. In Figure 2B, the fold induction of BRCA2 foci looks much more pronounced for WT than ∆DBD, but in the text, the authors state "difference in fold increase compared to cells producing full‐length BRCA2 was either small or absent". I don't see statistical analyses in the graph comparing the different samples to see if differences were significant, can the authors provide this?

10. There clearly seems to be more RAD51 foci in undamaged cells in the ∆DBD compared to WT BRCA2 (Figure 2D), can the authors highlight and comment on this?

11. Figure 2E, do RAD51 foci persist at 48 and 72 hrs in the deletion mutants?

12. I would change the title of this section "DBD and CTD are not essential for BRCA2 and RAD51 focus formation" (p. 9) to "The DBD and CTD are not essential for BRCA2 and RAD51 foci formation, however, the kinetics of recruitment are delayed" as quantified in Figure 2EandG.

---

## [Author Response]

Essential Revisions:The reviewers agree that this is an important, technically challenging and generally expertly executed study. As you will see from the individual reviewers' comments, the following revisions will be required to gain full confidence in the reported claims:1. Figures 1 and 2 need improvements with biological replicates and statistical tests. The data presented in these figures are promising, although would benefit from three biological replicates.

We apologize for not having included sufficient data replicates in our original submission. We have performed additional replicates of the cell survival experiments and foci counting experiments in figures 1D-F and 2B-C. The figures and legends have been revised to include the additional data. Statistical test results are included in the figure legends, main text and Source Data Files accompanied with Figures 1-3. The overall results are the same and the conclusions from them do not change. We agree that this strengthens our work and was a necessary improvement.

2. After extensive discussion, all three reviewers had concerns with interpretation of the RAD51 foci data. Figure 2 shows that RAD51 foci formation kinetics is slower in all BRCA2 truncation variants compared to full-length BRCA2 expressing cells. The reviewers felt that the authors somewhat over-interpreted their observations that RAD51 foci are eventually formed in all variants. The observed alteration in the number and intensity of foci at the early time points should be discussed.

We agree that the foci data is not absolute and in general this type of data is subject to variability from experiment(er) to experiment(er), not least of which is the difficulty of defining a focus (size, intensity, etc.) from different staining and microscopy techniques and different image analysis methods. However, our research group has years of experience analyzing foci of homologous recombination and other repair proteins. From this we observe that changes in foci quantitation (number of foci) within one set of experiments (same reagents/conditions used for cell treated and prepared for analysis) is a robust read out. Our point is that the number of foci does increase after irradiation for all cells lines tested 2 hours after radiation (WT p=6.09E-33; ∆DBD p=9.35E-03; ∆CTD p=1.98E-29; ∆DBD∆CTD p=1.28E-37), whereas in the cell lines lacking DBD it takes longer to reach the maximum number of foci.

The difference in foci number and intensity between the cell lines expressing the different BRCA2 variants is interesting and worthy of discussion. Cells expressing the BRCA2 variants lacking the DBD have more RAD51 foci before treatment compared to cells expressing full length BRCA2. This may reflect the importance of this domain in replication associated functions of BRCA2. Though the cells grow normally with a similar cell cycle profile (Figure 2 -Supplement 1) the role of BRCA2 underpinning replication may be slightly altered in these cell lines. Though we have not yet investigated this further we have added this observation to the discussion page 20, line 368.

3. SDS gels of purified constructs should also be included.

We apologize for this omission. We present coomassie and silver stained gels of our protein preps in Figure 4 – supplement 4. As noted by the reviewers, BRCA2 is difficult to purify and yields are low. Fortunately, only small amounts of protein are needed for SFM imaging. The 2 step purification we employ results in some contaminants in our final preps. From the silver stained gel it is evident that purity of the protein preps is similar and pattern of contaminants is the same in all. We do observe consistent results in the SFM analysis, volume and solidity, for multiple independent preparations of BRCA2, both here and in our previously published work. The vast majority of contaminants present are filtered out from the SFM analysis based on their size compared to the ~450 kD 2XMBP-BRCA2 proteins we are analyzing. Thus we are confident that the analysis represents BRCA2 protein behavior and differences are due to the introduced truncations.

4. There seems to be a discrepancy between the impact of structural changes in CTD truncated BRCA2 (shown in Figures4 and 5) and observed cellular phenotypes (Figure 1. MMC and Cisplatin resistance). The authors should elaborate their discussion on this.

This is an intriguing aspect of our results. We have added discussion of this aspect (Page 22, line 397). We agree that the relationship between our various assays is not simple. This points to complex functions of BRCA2 that are only beginning to be revealed and understood. Because the assays are very different, involving cells with all interacting components available vs. individual isolated proteins, we cannot at this point directly relate the protein structural changes to precise biological functions. However, we do note that the ΔDBDΔCTD and ΔDBD cells lines behave similarly and are both sensitive to MMC and Cis Pt. Purified ΔCTD is deficient in structural response, while the similar variant in cells is not sensitive the DNA crosslinking agent. All deletion variant proteins are defective in response to ssDNA while again the ΔCTD in cells is not overly sensitive to DNA crosslinking agents. Thus, we observe structural transition defects in all c-terminal deletion mutants while only those variants missing the DBD are sensitive in cell assays probing the function of BRCA2 in DNA cross link repair. We and others (Le et al., 2020) observe complex interactions between different parts of BRCA2 with itself (inter = multimerization and intra = conformation molecularly), that can be modulated by binding partners, including DSS1. Although important and interesting, including DSS1 interaction is outside of the scope of our current study. We continue to investigate the structural response of BRCA2, these and other variants, to additional binding partners and hope that these studies will eventually contribute to a clearer connection between protein conformational changes and biological functions.

Reviewer #1:[…]The major strength of this work lies in their comprehensive analyses of BRCA2 variants using a wide range of state-of-the-art in vivo and in vitro techniques, allowing straightforward comparison of their impact on cellular function, molecular behaviour and structural changes. The limitations of this study, although minor for the conclusion drawn by this study, are (1) CTD deletion generally confers modest cellular phenotypes compare to DBD deletion and is fully resistant to MMC and cisplatin. It remains unknown why CTD deletion elicits less impact despite its strong impairments in ligand-induced conformational changes;

This intriguing observation is more explicitly discussed as noted above for essential revision 4.

and (2) the molecular behaviours of BRCA2 in mouse ES cells might not be directly translated to these in human somatic cells.

It is of course possible that some aspects of BRCA2 behavior in human somatic cells and mouse ES cells differ. At least for diffusive behavior we have shown in our previous work (Reuter et al., J. Cell Biol. 2014, in manuscript reference list) that BRCA2 behaves the same in HeLa cells as in mouse ES cells.

My specific comments on each experimental data are outlined below:(1) Survival assays of respective mES cell lines show that CTD is important for normal resistance to IR and olaparib, but not for MMC or cisplatin, while DBD is important for all aforementioned treatments. Their analysis of HR competency, inferred by Cas9-induced gene targeting efficiency, revealed that the deletion of DBD, and of CTD to a lesser extent, impact on efficient integration of the reporter, concluding that these domains are important for HR repair of two-ended DSB. These results are robust and convincing.

(1) Thank you, does not need response.

(2) They then moved onto the analyses of the IR-induced RAD51 and BRCA2 foci formation. Surprisingly, they found that the deletion of DBD or CTD did not drastically affect foci formation, albeit slightly less efficient compared to full-length BRCA2. While the results and trends look promising, the number of samples analysed is somewhat limited (i.e., two or three technical replicates, rather than biological replicates) and the statistic tests have not been conducted.

Statistical tests are in the source data files as indicated in the figure legend.

For all cellular assays independent experiments have been performed at different days with cells at different passage numbers. Within all independent experiments we have included technical replicates (cell survivals: 2 or 3 wells; HR assays: 2 wells; microscopy experiments: at least 3 field of views per condition). To further support our observations, we have generated the single ΔDBD and ΔCTD cell lines and the cell line lacking both domains (ΔDBDΔCTD). Although in the original version of the manuscript we have included the results of statistical tests in the Source Data files, we have included additional information in the text and figure legend where appropriate.

(3) HeloTag also allowed them to assess the mobility of these BRCA2 variants in mES cells, using single-particle tracking (SPT). Focusing on S-phase cells, they show that the increase of the immobile fraction of BRCA2, detectable at 2-4 hours upon ionising radiation, is not severely affected by the deletion of DBD or CTD. The conclusion was drawn from the datasets from two independent experiments of at least 15 cells and ~10,000 tracks per condition, which, in my opinion, is respectful.

(3) Thank you, does not need response.

(4) Equivalent human BRCA2 deletion variants were purified from human HEK293 cells and subjected to scanning force microscopy (SFM) imaging. This analysis revealed that, while full-length BRCA2 commonly forms large oligomers of more than four molecules (70%), all the truncation variants showed somewhat reduced capacity to form tetramers or larger oligomers (i.e. ~44-54%). Upon RAD51 incubation, the majority of full-length BRCA2 (74%) became monomeric, while the C-terminal deletion appeared to respond less, with 40-55% becoming monomers and 30% remaining as dimers. The addition of ssDNA made full-length BRCA2 structure extended but elicited no structural impact on the truncated variants. These conclusions were drawn from the analysis of ~260-500 particles per sample, and look to me, credible. It would nevertheless be good to see the quality of purified BRCA2 variants by silver staining or mass-spectrometry to eliminate potential complications associated with other co-purified factors.

(4) A silver stained gel of the proteins used has been added to supplementary figures (Figure 4 – supplement 4) and this issue is addressed in essential revision number 3.

Line 61 “remarkable rearrangement by RAD51, ssDNA and ssDNA” – duplication of ssDNA.

Line 61 duplicated text has been removed.

Line 79 'phosphorylation-dependent, RAD51 interaction domain' – RAD51 interaction with CTD is 'blocked' by the phosphorylation at S3291. The sentence might be misread as 'CTD phosphorylation promotes RAD51 interaction'. Make this clear at least for the first time when it is refereed.

Line 79 We agree this sentence should be phrased differently. We have adjusted it to this:

“RAD51 interaction domain at the C-terminus of BRCA2 which is inhibited by cell-cycle regulated BRCA2 phosphorylation (Esashi et al., 2005).”Figure 1 – Colour codes do not always match. For example, in panels A and G, δ-DBD is shown in blue, but in panels C-F, it is shown in purple. It will be easier to read the results if they are shown in consistent colour codes.

Figure 1 color codes have been adjusted in panels C-F for consistency.

Figure 2 – This figure was hardest for me to understand due to three main issues. Firstly, while the manuscript discusses the kinetics of foci formation at any given time point, the results are clustered according to the type of samples (i.e. full-length, dDBD, dCTD and dDBD/dCTD), rather than each time points. Clustering treatment or time point (i.e. with and without IR, or hours post-IR) would make it easier to understand. Secondly, statistic tests had not been conducted, hence it is difficult to assess the conclusions are supported by their results. In fact, the deletion mutants seem to show significant difference from full-length BRCA2 at 2 hour after IR. Finally, as they are assessing the action of BRCA2 in the context of RAD51 recruitment, they could include the quantitative assessment of BRCA2-RAD51 co-localising foci. This can be done by ImageJ co-localisation plugin.

Figure 2 Concerning the choice to cluster results by cell line vs by time point; our focus was to examine the response of each cell line to compare to each other. In our opinion the presentation as in figure 2 shows the pattern of response per cell line best. The reviewer’s suggestion is however also a good way to display differences and we have added this presentation in the supplemental figure (Figure 2 – Supplement 1).

Figures 4 and 5 – These results are well presented in my opinion. The manuscript, however, goes back and forth to explain these datasets, i.e., first explains BRCA2 oligomer formation (page 15), followed by solidity of BRCA2 affected by ssDNA (page 16), then back to BRCA2 oligomer status affected by RAD51 (page 17). It will be more straightforward if the data shown in Figure 4 are explained fully, before moving to the solidity assessment shown in Figure 5.

Figures 4 and 5 We thank the reviewer for the noting that the text does not follow the figures in order of their presentation. Although we had our reasons for this order it is not essential to understanding and can, as pointed out, be confusing. We have re-ordered the text to place all description of results in figure 4 before description of results in figure 5.

The beauty of this study is its consistency in the samples analysed. It will be good to include a table, summering the phenotypes and molecular properties of each BRCA2 variants.

Thank you for this suggestion, we have included a table summarizing the in vivo and in vitro observations in this study (Table 1).

Reviewer #2:[…]1) Figure 1 and 2: It is a standard in the field that cellular experiments presented should have at least three biological replicates for chlonogenic survival assays. Please perform the biological and technical replicates for data presented in Figure 1 and Figure 2.a. Figure 1:i. More than two concentrations of (Olaparib) should be assayed in chlonogenic survival to understand the full curve of cellular survival.ii. Presentation of the statistical analysis directly in the Figure C-F would be helpful in ascertaining importance with a description in the manuscript.iii. Displaying the chlonogenic data in a non-logarithmic form would be helpful to viewers for comparing the IC50 of survival.iv. The description of how cellular sensitivities to IR, Olaparib, MMC, and cisplatin relate to repair outcome is lacking. Please describe in more detail how each agent causes damage and the potential outcomes and DNA repair pathways that would be utilized for repair.

(a) Figures 1 and 2. Additional replicate experiments have been added as suggested, see also response to essential revisions. We apologize for our mistake leaving off the highest concentration of Olaparib. For consistency we have repeated those experiments n=3 with 3 different doses. The outcome of the experiment was similar and did not alter our conclusions. Statistical analysis can be found in the table in supplemental material (Figure 1 and 2 – Source data files).

In our experience (such assays are standard in our department for more than 30 years) log scales are expected and preferred. Log scale better shows the level of sensitivity of the cell lines in a space efficient way (hope you appreciate humor – https://xkcd.com/1162/).

We have included additional information on BRCA2 function in relation with the different DNA damage response induced by the reagents used (Results, line 117-122).

(b) Address biological replicates. Statistical analysis provided in Figure 1 – Source data file 1 concerning the role of c-terminal DBD in Rad51 turnover, the possibility we favor is that BRCA2 conformational changes are an important aspect of its function and that the absence of the DBD changes this. This concept is the topic of the last paragraph in the discussion and the last sentence of the paragraph before that. It happens all too often that statement in the discussion of an article are referenced as fact and thus incorrectly propagated in the literature. We leave further speculation and inspiration for new experiments up to the readers and discussions among colleagues.

b. Figure 2: While the DBD and CTD are not essential for BRCA2 and RAD51 foci formation the data presented in Figure 2E suggest that the DBD plays the most important role. The kinetics of RAD51 foci formation/behavior of the δ DBD shows a different behavior compared to full length BRCA2 and δ CTD.i. It is unclear if this phenomena is different and could benefit from increased biological replicates and display of error. Statistical analysis for Figures 2B,D,C, and F would be helpful in ascertaining importance.ii. Moreover, how the authors hypothesize this domain plays a role in RAD51 foci turnover would be helpful as this domain does not directly interact with RAD51 protein.c. A more mechanistic functional role of these domains can be obtained by single-molecule analysis of BRCA2 and the deletion constructs with varying substrates and BRCA2 binding partners.i. It is unclear why the CTD is not sensitized to MMC or Cisplatin but previous work in the field has shown a role for this domain in fork protection and crosslink repair (Atanassov et al., etc) as the authors describe in the discussion.

(i) Concerning the lack of MMC and Cis-Pt sensitivity of ΔCTD. It is indeed correct that several papers in the past have indicated the relevance of the CTD for crosslink repair (Atanassov et al., 2005; Donoho et al., 2003; Marple et al., 2006). As indicated in our discussion (page 21, line 388) the different observations in those study could possibly be accounted for by differences in the BRCA2 alleles and cell types (MEFs) that were used.

ii. These discrepancies and a mechanism may be discerned by evaluating single-molecule analysis with DSS1 (known DBD interaction partner) and fork substrates with and without crosslinks. Other protein binding partners like DSS1 and p53 etc. may be important for dissecting out function of the studied domains. These revisions should be easily completed by the talented authors within 6 months.

(ii) We agree that interaction with DSS1 and possibly other binding partners is of great interest for our and similar work. Independent of the assumptions as to the quality and availability of our laboratory personnel, this is not the focus of our current work and outside the scope. We are happy that the reviewer gains such inspiration from our results. We would in general argue that delaying publication of this, or any well done and presented, work for 6 months or longer does not serve the greater scientific community.

1. For p53: Rajagopalan S. et al., PNAS 2010 PMID: 20421506

Figure 3 has been adapted to clarify the meaning of the data variance presented. The values below the plot were the mean values while in the boxplot median and 25 and 75% of the data are indicated. Median values are now indicated below the plot.

3. The manuscript would benefit from a figure which depicts a model for the function of each domain in BRCA2 and how the findings of these domains lay the foundation for the field.

We believe the reviewer is referring to protein ball – DNA lines sort of cartoons. If that is the case, we strongly disagree. We feel that often these cartoons imply information / interactions / specific functions that are simply not yet known or supported by data. These elements may not be the artists intention or their main point, but they are often interpreted as scientific fact. We think such cartoons should be avoided and if used carefully annotated as to what parts are based on consensus data and what parts are drawn because steps or interactions have to be filled in.

Reviewer #3:[…]Strengths:1. Generation of mouse ES cells with both endogenous alleles of BRCA2 containing the deletion mutations in addition to a HALO tag is an incredible technical breakthrough and will be a highly valuable reagent for genetic and cell biological studies of mouse BRCA2.

1. Concerning comparison of cell sensitivity of our BRCA2 deletion variants and “completely non-functional BRCA2 allele”; This is indeed a good idea and would be interesting to pursue. However, we note that this would require making specific mutations from the human protein in mouse ES cell lines and thus require possibly substantial work determining if they mutations behave the same of differently. Although cell lines expressing (patient derived and other) BRCA2 truncations and deletion variants are described as “completely non-functional” this description does not entirely make sense to us. Cells lacking an essential protein (BRCA2) are, we assume by definition, dead or dying. That some tumor derived cell lines survive with apparently severe BRCA2 defects may attest to their other genetic alterations. A “clean” comparison in mouse ES cells does not exist. For our survivals in mouse ES cells we used a RAD54 deletion cell line as a well characterized comparison as HR defective in response to ionizing radiation. Though not perfect this at least provides a means of comparing sensitivity (Figure 1C) where the two BRCA2 deletion variants are even more sensitive.

2. The deletion mutants ablating either the DBD or the CTD, or both, is a great genetic approach to understanding the role of these key domains in BRCA2. The response of these mutants (versus WT BRCA2 as a benchmark) to various DNA damage (IR, PARPI, MMC, cisplatin) provides interesting information delineating the roles of these two important domains in BRCA2. For example, the ∆CTD mutant is significantly sensitive to IR and Olaparib, yet complements as well as WT BRCA2 in response to the crosslinking agents MMC and cisplatin.

2. Concerning mechanistic importance (insight) from SPT analysis. The function of BRCA2 and other DNA repair proteins logically require them to become localized/temporary immobile at sites of damage where they need to exercise biochemical activities. This is seen as a high local concentration in “foci”. In order to accumulate in this way or simply become localized to do its work a protein has to change its diffusive behavior, either more of the protein moves to / through a place or more of it stay immobile for a longer time. This is what we can quantify by SPT. Here we show that, perhaps contrary to expectations, the in vitro defined DNA binding domain is not required for this immobilization or change in diffusive behavior. This lack of effect could be described as a negative result, however just as important to communicate and valid as if we had detected an effect. We discussed the mechanistic implication and motivation for SPT study of BRCA2 in a previous publication (Reuter et al., JCB, 2014 in the reference list of our current manuscript). There we also explain how the number of proteins that change mobility and the magnitude of their change in mobility is consistent with the expected amount of damage inflicted.

3. The BRCA2 protein is notoriously difficult to purify and yet the authors succeeded in purifying 4 different forms of the protein for biophysical analysis. While it is difficult to interpret the various forms of BRCA2 by SFM, there are clear differences in the architecture between WT and the three c-terminal mutants. These differences are highlighted upon binding to RAD51 or ssDNA.General Comments:It is unclear how missing the c-terminal domain (CTD) or the DNA binding domain (DBD) of BRCA2 can be interpreted as having "roles beyond delivering strand exchange protein RAD51" unless a complete biochemical workup of the deletion mutants was performed to detect any alterations in DNA binding, stimulation of RAD51 dependent strand exchange, etc… While interesting and certainly an impressive technical feat, foci imaging and single particle tracking do not provide much information on mechanism (i.e. whether BRCA2 is binding DNA and loading/nucleating RAD51).The interpretations in the discussion are not overstated, however, I somewhat disagree with the notion that the data, as presented, clarifies the role of BRCA2 beyond its canonical functions of RAD51 loading and nucleation on resected DNA substrates. I would have liked if the authors discussed the idea that it is surprising that mouse ES cells can tolerate complete loss of the DBD, CTD, and loss of both together. Questions that should be addressed in include some of the following: Are proliferation rates compromised compared to WT cells?

We did not observe compromised growth rates compared to WT cells. We have included this observation in the results (page 7, line 110).

Are they experiencing replication stress in the absence of any exogenous damage?

The difference in number of spontaneous RAD51 foci we observe in untreated cells lacking the DBD could be an indication for increased replication-associated DNA damage. This interesting topic is ongoing work of a departmental collaborator and hence is here. We have however highlighted this observation in the discussion (page 20, line 368).

Further, is there something unique about mouse ES cells that may differentiate BRCA2 behavior that would be expected in somatic human cells?It is interesting to note that many years ago Ashworth and Taniguchi published back-to-back papers in Nature (2008) describing BRCA2 reversion alleles from in vitro screens of BRCA2 mutant cells selected in cisplatin or PARPi such that some of these reversions resulted in huge deletions of the entire DBD of BRCA2, and yet, they promoted resistance to PARPi. In this context, I would much appreciate if the authors commented on their findings that their constructed DBD deletion is not resistant to PARPi and if they offered some speculation as to why the reversions in those previous studies were.1. Are the BRCA2 mouse alleles missing the c-terminus (∆CTD) properly localized to the nucleus? The authors do mention that mouse BRCA2 contains an NLS on the N-terminus of the protein but did they check? The human BRCA2 ∆CTD protein would be expected in the cytoplasm as NLS are located at the extreme c-terminus of the protein.

As can be appreciate from Figure 2A all variants that we studied localize to the nucleus. Consistent with our observations, others have also observed N-terminal fragments of murine BRCA2 localizing to the nucleus (Sarkisian et al., 2001).

2. In the figure 1A schematic, can the HALO tag graphic be placed on the c-terminus of the three deletion mutants? It might be confusing left only on the full length BRCA2 depiction.

The HaloTag is only indicated for the full length as the deletion mutants are shown for both the in vivo experiments in ES cells as for the in vitro experiments with N-terminally MBP-tagged purified BRCA2. We think it would be more confusing to add all tags and thus need to explain multiple versions as the purpose of the diagrams is to indicate the location of the deletions.

3. Do the three purified human BRCA2 proteins bind DNA substrates in an EMSA experiment? It would be informative to know if binding to certain substrates is diminished or ablated, for example, on ssDNA vs dsDNA vs 3' and 5' tails.

We agree this would be interesting to know but do not know how biologically significant it would be given that we do not see evidence of diminished immobilization, presumably DNA bound, at sites of damage in cells. Nonetheless it is not practical for us to do EMSA’s because our yields of BRCA2 are low in amount and concentration. We would not expect to see a good result in EMSA, in any case we could not do titrations for proper quantification.

4. A similar question is if the purified mutants are compromised for binding RAD51, and more specifically, binding and stabilizing the RAD51 filament as previously demonstrated for the CTD (Esashi et al. 2005 and 2007, Davies et al. 2007)?5. In the first paragraph of page 5 (lines 74-86), it would be helpful if the authors discussed briefly the separation-of-function of the BRC repeats in BRCA2 (Carreira et al. 2011, Chatterjee et al. 2016) especially in the context that BRC5-8 seems to operate in a mechanistic manner similar to the CTD in stabilization of the RAD51 filament on ssDNA.

The work referred to is indeed quite interesting. There is considerable work (also by our group) on isolated bits of BRCA2, some of it contradictory and in our opinion some of it possibly not representative of domain function in the context of the complete protein. Filament formation and stability are not the addressed in this current work so we choose not to discuss this topic and the associated references.

6. Figure 1C-E: please increase size of data labels (boxes, circles, triangles) and lines in the survival curve graphs to make it easier for readers to visualize the results.7. In the survival assays in Figure 1 C-E, why didn't the authors include a classical BRCA2 pathogenic mutation (or null) to use as a benchmark for comparison to ∆DBD and ∆CTD? Inviable? It would be a very informative comparison if possible.

BRCA2 null mouse ES cells are not viable and comparison of survival with other cell lines will be difficult. Please see our response to Public Review Weakness above; Concerning comparison of cell sensitivity of our BRCA2 deletion variants and “completely non-functional BRCA2 allele”.

8. Can the authors comment on the 2 bands in Figure 1B western confirming expression of the HALO tagged mutant alleles in mouse ES cells, is one BRCA2 and the other a degradation product, or is one of the alleles truncated on the N-terminus? Were both alleles completely sequenced in the cells used for the studies?

Indeed we observe a double band of BRCA2 which is due to incomplete cleavage of the F2A peptide in the Halo-F2A-neomycin cassette. As can be appreciated from the survival plots that this incomplete cleavage does not appear to affect BRCA2 function as cells expressing the full length protein are not sensitive to IR compared to wild type mouse ES cells. Extensive genotyping of the knock in cell lines can be found in Figure 1 – supplement 1.

9. In Figure 2B, the fold induction of BRCA2 foci looks much more pronounced for WT than ∆DBD, but in the text, the authors state "difference in fold increase compared to cells producing full‐length BRCA2 was either small or absent". I don't see statistical analyses in the graph comparing the different samples to see if differences were significant, can the authors provide this?

Statistics analysis is presented in the supplementary data file and referred to in the figure legends. We agree with the reviewer that the statement is somewhat confusing and rephased:

“Upon irradiation, the number of BRCA2 foci increased in all deletion variants, (p<0.001 for all BRCA2 variants). However total number of BRCA2 foci appeared lower after radiation in all deletion variants.”

10. There clearly seems to be more RAD51 foci in undamaged cells in the ∆DBD compared to WT BRCA2 (Figure 2D), can the authors highlight and comment on this?

As indicated above we have include this observation in our discussion (page 20, line 368).

11. Figure 2E, do RAD51 foci persist at 48 and 72 hrs in the deletion mutants?

We have not tested this, in general ES cells divide every 24 hours so the time points suggested would be difficult to compare. As ∆DBD cells are sensitive to IR the percentage of surviving cells is much smaller that their FL counterparts which will grow to much higher density, this would make it difficult to quantify foci in those nuclei.

12. I would change the title of this section "DBD and CTD are not essential for BRCA2 and RAD51 focus formation" (p. 9) to "The DBD and CTD are not essential for BRCA2 and RAD51 foci formation, however, the kinetics of recruitment are delayed" as quantified in Figure 2E and G.

We appreciate the suggestion to qualify this but find the longer version cumbersome and not as effective as a heading. To better reflect the results in the section we change the heading to: “The DBD and CTD affect BRCA2 and RAD51 focus kinetics”

References

Atanassov, B. S., Barrett, J. C., and Davis, B. J. (2005). Homozygous germ line mutation in exon 27 of murine Brca2 disrupts the Fancd2-Brca2 pathway in the homologous recombination-mediated DNA interstrand cross-links' repair but does not affect meiosis. *Genes Chromosomes Cancer, 44*(4), 429-437. doi:10.1002/gcc.20255

Donoho, G., Brenneman, M. A., Cui, T. X., Donoviel, D., Vogel, H., Goodwin, E. H., Chen, D. J., and Hasty, P. (2003). Deletion of Brca2 exon 27 causes hypersensitivity to DNA crosslinks, chromosomal instability, and reduced life span in mice. *Genes Chromosomes Cancer, 36*(4), 317-331. doi:10.1002/gcc.10148

Le, H. P., Ma, X., Vaquero, J., Brinkmeyer, M., Guo, F., Heyer, W. D., and Liu, J. (2020). DSS1 and ssDNA regulate oligomerization of BRCA2. *Nucleic Acids Res*. doi:10.1093/nar/gkaa555

Marple, T., Kim, T. M., and Hasty, P. (2006). Embryonic stem cells deficient for Brca2 or Blm exhibit divergent genotoxic profiles that support opposing activities during homologous recombination. *Mutat Res, 602*(1-2), 110-120. doi:10.1016/j.mrfmmm.2006.08.005

Sarkisian, C. J., Master, S. R., Huber, L. J., Ha, S. I., and Chodosh, L. A. (2001). Analysis of murine Brca2 reveals conservation of protein-protein interactions but differences in nuclear localization signals. *J Biol Chem, 276*(40), 37640-37648. doi:10.1074/jbc.M106281200